# The ng_ζ1 toxin of the gonococcal epsilon/zeta toxin/antitoxin system drains precursors for cell wall synthesis

Andrea Rocker[1], Madeleine Peschke[1], Tiia Kittilä[1], Roman Sakson[1], Clara Brieke[1] & Anton Meinhart[1,2]

Bacterial toxin–antitoxin complexes are emerging as key players modulating bacterial physiology as activation of toxins induces stasis or programmed cell death by interference with vital cellular processes. Zeta toxins, which are prevalent in many bacterial genomes, were shown to interfere with cell wall formation by perturbing peptidoglycan synthesis in Gram-positive bacteria. Here, we characterize the epsilon/zeta toxin–antitoxin (TA) homologue from the Gram-negative pathogen *Neisseria gonorrhoeae* termed ng_ε1 / ng_ζ1. Contrary to previously studied streptococcal epsilon/zeta TA systems, ng_ε1 has an epsilon-unrelated fold and ng_ζ1 displays broader substrate specificity and phosphorylates multiple UDP-activated sugars that are precursors of peptidoglycan and lipopolysaccharide synthesis. Moreover, the phosphorylation site is different from the streptococcal zeta toxins, resulting in a different interference with cell wall synthesis. This difference most likely reflects adaptation to the individual cell wall composition of Gram-negative and Gram-positive organisms but also the distinct involvement of cell wall components in virulence.

[1] Department of Biomolecular Mechanisms, Max Planck Institute for Medical Research, Jahnstr. 29, 69120 Heidelberg, Germany. [2] Present address: Research Institute for Molecular Pathology (IMP), Vienna Biocenter (VBC), Campus-Vienna-Biocenter 1, 1030 Vienna, Austria. These authors contributed equally: Andrea Rocker, Madeleine Peschke. Correspondence and requests for materials should be addressed to A.M. (email: Anton.Meinhart@imp.ac.at)

Bacterial toxin–antitoxin (TA) modules are poisonous gene pairs coding for a toxin and its cognate antitoxin. TA systems are highly prevalent in bacterial genomes[1,2] and in most cases the neutralizing antitoxin is encoded together with the toxin on a bicistronic locus[3,4]. TA modules were initially discovered as plasmid stabilizing systems, but have since been shown to regulate persister cell and biofilm formation, to act as stress response elements and even to increase virulence of pathogenic bacteria[5–7]. Because of their roles in bacterial survival, TA systems have become interesting targets for the development of new antimicrobial agents. Detailed understanding of TA systems is, however, required before such approaches can be efficiently utilized.

TA modules can be grouped into six different types varying in their mode of toxin inhibition[8]. Typically, the toxin is a protein whereas the antitoxin can be either a protein or a non-coding RNA. The best characterized family of TA systems are the type II systems where both the toxin and the antitoxin are proteins[3,9]. Neutralization of the toxin is achieved in type II systems through complex formation between the two proteins. Regardless of the TA system type, toxins interfere with vital cellular pathways leading to cell death or dormancy[10]. Most toxins known to date target protein translation, e.g., by cleavage of mRNAs[11–13], modification of tRNAs,[14,15] or inhibition of ribosome activity[16]. However, toxins have also been shown to inhibit replication[17,18]

and to prevent ATP synthesis by pore formation[19,20]. In addition, one group of type II toxins, the so called zeta toxins, interfere with cell wall integrity by phosphorylating cytosolic cell wall precursors[21] and are highly prevalent in pathogenic bacteria[22].

Zeta toxins are canonical type II TA systems in which the toxicity of zeta is inhibited by the epsilon antitoxin. The first characterized members of this family were the plasmid encoded ε/ζ from *Streptococcus pyogenes*[23] and the chromosomally encoded PezAT systems from *Streptococcus pneumoniae*[24]. Zeta toxins adopt a classical phosphotransferase fold[24,25] with a central P-loop motif important for ATP binding[26]. In contrast, the cognate epsilon antitoxin folds into a simple three helix bundle domain. Both streptococcal systems form an epsilon$_2$zeta$_2$ heterotetramer, in which the ATP binding to the active site of zeta is blocked by epsilon impairing zeta activity. In fact, zeta toxins use ATP to phosphorylate UDP-*N*-acetylglucosamine (UNAG) to form UDP-*N*-acetylglucosamine-3′phosphate (UNAG-3P)[21] and thereby modify this essential peptidoglycan precursor. Biosynthesis of peptidoglycan is highly conserved in bacteria and requires a cascade of enzymes that sequentially modify UNAG[27]. Initially, an enolpyruvate group is added to the C3′-OH group of UNAG by MurA yielding UDP-acetylglucosamine enolpyruvate (EP-UNAG). Because zeta toxins and MurA act on the same C3′-OH group, formation of UNAG-3P inhibits formation of EP-UNAG. When zeta is inactive, e.g., by binding to the epsilon antitoxin, the

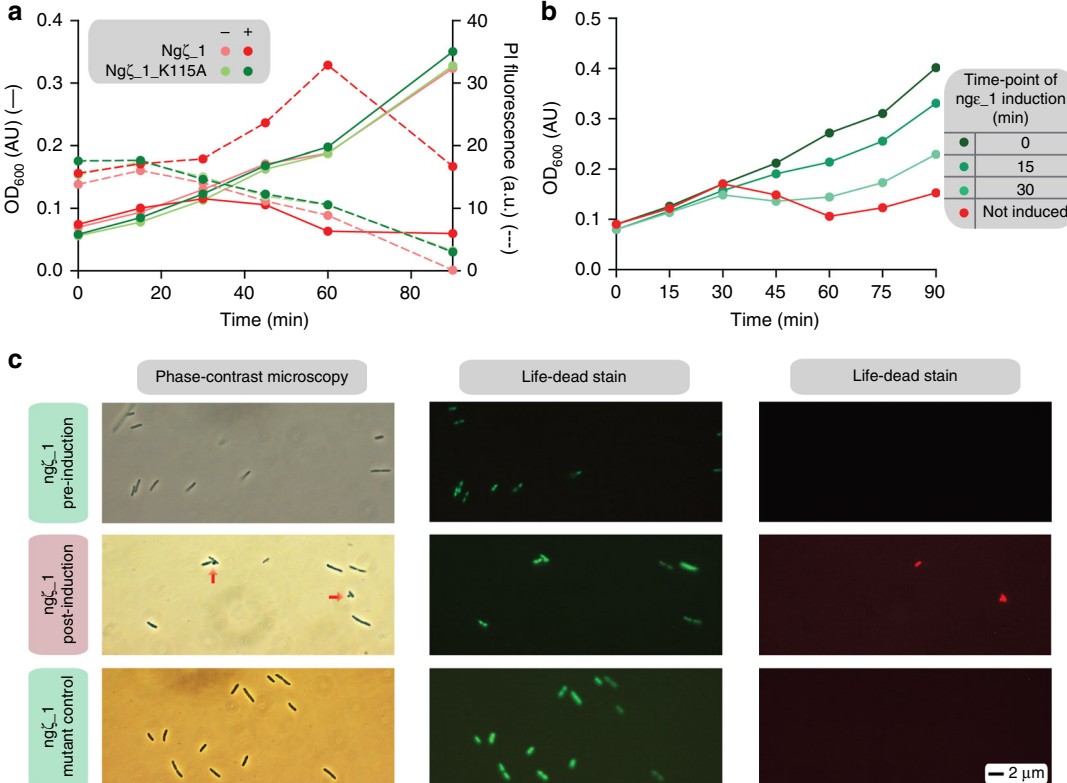

**Fig. 1** Ngζ_1 expression in *E. coli* cells causes a lytic phenotype compensated by ngε_1 co-expression. **a** Growth curves of *E. coli* C41(DE3) cells expressing ngζ_1 (red solid line and left *Y*-axis) or ngζ_1(K115A) (forest-green solid line and left *Y*-axis) together with the corresponding propidium iodide influx into the cells (dashed lines similarly colored and right *Y*-axis). Control experiments in which expression of ngζ_1 or ngζ_1(K115A) was not induced are shown in light pink and lime-green. Note that growth curves of the two uninduced cell cultures as well as the PI measurements of the mutated variant (induced and not induced) fully overlay with each other. **b** Growth curves of *E. coli* BL21(DE3)-RIL cells co-expressing ngζ_1 and ngε_1. Expression of ngε_1 was induced 0 (forest-green), 15 (lime-green) and 30 min (pale-green) subsequent to ngζ_1 induction. The control experiment in which expression of ngε_1 was not induced is shown in red. **c** Phase contrast and fluorescence microscopy (live-dead stain) of *E. coli* C41(DE3) cells before and after ngζ_1 induction (30 min) and the control experiment using the inactive ngζ_1(K115A) mutant. Cells with green fluorescence have intact cell membranes and are alive. Membrane permeable cells have red fluorescence due to propidium iodide influx. Resolved bulges due to cell wall extrusions in phase-contrast microscopy images are highlighted with red arrows

enolpyruvate moiety of EP-UNAG is reduced by MurB yielding UDP-muramic acid (UNAM), the activated sugar to which a peptide stem is attached. Importantly, UNAG-3P is not only a dead-end product of zeta toxins but acts as a potent inhibitor of MurA[21]. Eventually, the combined types of MurA inhibition stall de novo peptidoglycan synthesis at this early step and thus cause cell lysis in rapidly dividing bacteria.

Only epsilon/zeta modules of Gram-positive bacteria have been characterized so far in detail, but homologs are also highly prevalent in Gram-negative bacteria[22]. In contrast to zeta toxins from Gram-positive prokaryotes, the few investigated homologous TA systems of Gram-negative bacteria seem to be much more diverse in their function and mechanisms. For instance, the structurally homologous plant effector protein AvrRxo1 from Xanthomonas[28] functions as a bacterial TA-system as well[29]. AvrRxo1 has been shown to phosphorylate NAD and its precursors NAAD in vitro and causes massive 3′-NADP accumulation in vivo[30,31]. On the other side, a zeta homologue that is prevalent in *Escherichia coli* strains combines toxin and antitoxin functionalities in a single polypeptide chain[32].

A prototype of a Gram-negative epsilon/zeta TA module, the epsilon_1/zeta_1 (ngε_1/ngζ_1) TA system, is encoded by *Neisseria gonorrhoeae*, an obligate human pathogen causing the sexually transmitted disease gonorrhoea[33]. Traditional antibiotic treatment of gonorrhoea consists of ampicillin and tetracycline but this therapy is threatened by the spread of antibiotic resistances. High-level tetracycline resistance is conferred by the *tetM* determinant of 25.2 MDa conjugative plasmids and can thereby rapidly spread across the population[34]. The tetM determinant was acquired by insertion into the genetic load region of smaller 24.5 MDa conjugative plasmids of the "Dutch" or "American" type[33]. Sequence analysis of the tetM-less ancestor as well as different 25.2 MDa plasmids revealed the presence of the *ngε_1/ngζ_1* locus in all plasmids[33]. Interestingly, all three different conjugative plasmids where shown to contain a second epsilon/zeta homologous locus that has diverged by extensive mutations between the 25.2 MDa *tetM*-containing and the 24.5 MDa conjugative plasmids. *TetM*-containing plasmids carry the *epsilon_2/zeta_2* locus, 24.5 MDa conjugative plasmids carry the *epsilon_3/zeta_3* locus[33]. Whereas the diverged epsilon_2/zeta_2 and epsilon_3/zeta_3 TA systems seem to be very similar in their amino acid sequence to the well characterized streptococcal systems (around 40% amino acid sequence homology), ngε_1/ngζ_1 seems to be very different even at this level. However, nothing is known about these two gene pairs and therefore the mechanistic relevance of two homologous systems on the same plasmid remained unclear so far. Thus, we set out to investigate the *ngε_1/ngζ_1* locus of *N. gonorrhoeae*. We could show that ngζ_1, like streptococcal zeta toxins, phosphorylates peptidoglycan precursors and efficiently inhibits cell wall synthesis. However, in contrast to the known zeta toxins, ngζ_1 showed topological rewiring and a hitherto unknown enzymatic activity by phosphorylating UDP-activated sugars at the C4′-OH group of the hexose moiety. This phosphorylation led to formation of dead-end metabolites resulting in cell death by draining precursors required for cell wall synthesis. As phosphorylation of the C4′-OH group has not been observed for the previously characterized epsilon/zeta systems, ngζ_1 and its proteobacterial homologs form a yet undescribed subfamily of zeta toxins.

## Results

**ngζ_1 forms a new subclass of zeta-like toxins.** Bioinformatic analysis of ngε_1/ngζ_1 revealed ngζ_1 toxins to be prevalent in numerous *N. gonorrhoeae* isolates (with currently 46 different NCBI entries with 100% sequence identity) but also a significant

number of closely related zeta homologs in more than 25 different, mainly pathogenic proteobacteria was found (Supplementary Fig. 1 and Supplementary Table 1). However, apart from being encoded on a bicistronic operon[33] and harbouring a P-loop motive, a hallmark for ATP/GTP binding proteins[26], ngζ_1 is remarkably different from the hitherto functionally characterized streptococcal zeta toxins[21,35] in its primary sequence. Especially striking is that the P-loop motive is located much closer to the C-terminus when compared with streptococcal zeta toxins (Supplementary Fig. 1). Furthermore, also the ngε_1 antitoxin has no similarities to any known epsilon protein. Thus, we first questioned, whether ngε_1/ngζ_1 is a functional TA system at all.

When ngζ_1 was expressed in *E. coli* cells, we observed a strong lytic phenotype 30 min post-induction as monitored by a decrease in optical density at 600 nm ($OD_{600}$) paired with an influx of propidium iodide (Fig. 1a). This toxic phenotype was suppressed

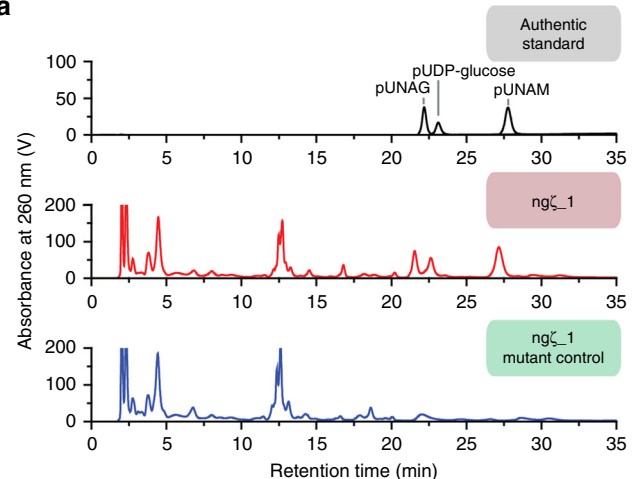

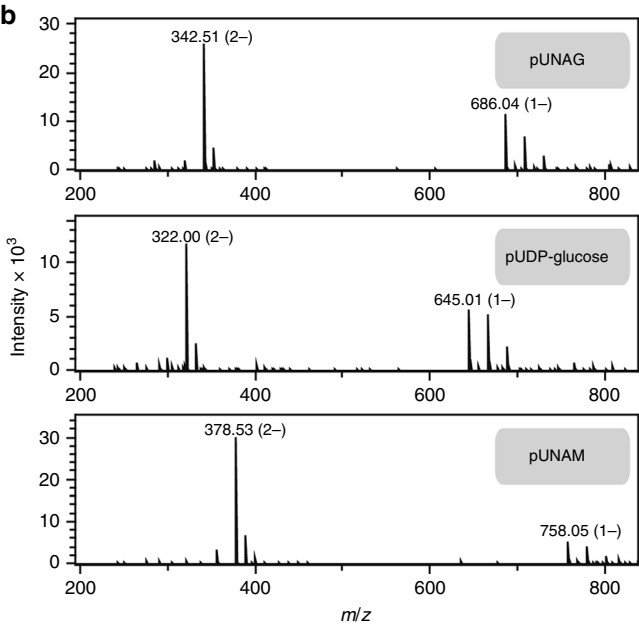

**Fig. 2** Ngζ_1 phosphorylates different UDP-activated sugar species. **a** High pressure liquid chromatography analysis of small metabolite extracts from *E. coli* strain C41(DE3) cells expressing either ngζ_1 (red) or the inactive ngζ_1(K115A) variant (blue). Separation of UNAG, UDP-glucose and UNAM phosphorylated by ngζ_1 in vitro are shown for genuine standards (gray). **b** Mass determination of isolated, phosphorylated UNAG, UDP-glucose and UNAM by ESI-MS

when ngε_1 was simultaneously expressed in *trans* (Fig. 1b) indicating that ngε_1 and ngζ_1 together are indeed a TA pair. However, induction of ngε_1 15 or 30 min after ngζ_1 was not sufficient to inhibit the lytic phenotype completely (Fig. 1b). Similar as reported for the pneumococcal zeta toxin[24], cell growth resumed in those experiments approximately 60 min after induction in all cases (Fig. 1b). Live/dead staining showed propidium iodide influx after induction of ngζ_1 and substantial cell wall defects of ngζ_1 expressing cells were observed (Fig. 1c). In order to provide evidence that a potential kinase activity of ngζ_1 causes the lytic phenotype, the catalytic lysine residue in the P-loop was altered to alanine (ngζ_1(K115A)). Cells expressing this variant grew similar to cells co-expressing ngζ_1 and ngε_1 (Fig. 1a, c). In conclusion, these findings strongly suggest that ngε_1/ngζ_1 is indeed an authentic TA operon and that the toxic effect of ngζ_1 is caused by its kinase activity.

**ngζ_1 phosphorylates peptidoglycan precursors.** We wondered whether similar to the streptococcal zeta toxins[21] ngζ_1 also phosphorylates UNAG. Thus, we isolated small metabolite extracts from cells expressing ngζ_1 or the catalytically inactive variant and separated these extracts by high performance liquid chromatography (HPLC) (Fig. 2a). Surprisingly, we observed significant accumulation of three new species exclusively in ngζ_1-poisoned cell extracts, a finding which sets ngζ_1 apart from the hitherto characterized zeta toxins that only showed UNAG-3P enrichment[21]. These three species were isolated and their masses were determined by ESI-MS (Fig. 2b). The eluting species had a molecular mass comparable to that of a phosphorylated, deprotonated modification of UNAG ($m_{obs.} = 686.04$ Da; $m_{calc.} = 686.04$ Da; first eluting species), of UDP-glucose/ UDP-galactose ($m_{obs.} = 645.01$ Da; $m_{calc.} = 645.12$ Da; second eluting species), and of UNAM ($m_{obs.} = 758.05$ Da; $m_{calc.} = 758.39$ Da, third eluting species). The second species is most likely phosphorylated UDP-glucose and not UDP-galactose since the *E. coli* strain C41(DE3) used in the experiment lacks the *galT* and *galE* genes[36,37] and is thus defective in UDP-galactose synthesis[38]. Detecting phosphorylated UNAM was quite surprising as the hitherto characterized zeta toxins phosphorylate UNAG at C3′-OH group[21], which is blocked by a lactoyl group in case of UNAM (Supplementary Fig. 2). These results strongly suggest that ngζ_1 phosphorylates UDP-sugars at a different site when compared with previously characterized zeta toxins.

To further verify the identity of these products, we tested the in vitro activity of ngζ_1, which was separated from ngε_1 chromatographically (Supplementary Fig. 3), for different UDP-sugars. In a coupled spectroscopic assay, UNAM strongly stimulated ATPase activity of the protein, but we could not detect significant amounts of ADP production when either UNAG or UDP-glucose were added (Supplementary Fig. 4a).

To account for low reaction rates, we incubated ngζ_1 with UDP-glucose (Supplementary Fig. 4b) or UNAG (Supplementary Fig. 4c) for up to 24 h and analysed the products by anion exchange chromatography. Indeed, phosphorylated products were detected in all cases. However, ADP production was found to be in excess over formation of phosphorylated UNAG when monitoring ngζ_1 activity over long periods of time. Surprisingly, the phosphorylated species vanished, when the assay was incubated for 12 h (Supplementary Fig. 4c) indicating a phosphatase activity of either ngζ_1 or an unidentified contamination. Ultimately, the three phosphorylated in vitro products were isolated and analysed by HPLC, where they had the same retention times as the species formed in vivo (Fig. 2a). Altogether, these results show that ngζ_1 phosphorylates several

| | ngε_1/ngζ_1 (K115A) Se-Met | ngε_1/ngζ_1 UNAM | ngε_1/ngζ_1 UNAM-4P |
|---|---|---|---|
| **Data collection** | | | |
| Space group | $P2_1$ | $P2_1$ | $P2_1$ |
| Cell dimensions | | | |
| $a, b, c$ (Å) | 79.8, 149.6, 125.1 | 80.3, 148.2, 124.4 | 80.5, 149.3, 125.4 |
| $\alpha, \beta, \gamma$ (°) | 90, 94.82, 90 | 90, 93.88, 90 | 90, 94.35, 90 |
| Resolution (Å) | 50–2.4 (2.8–2.4) | 50–2.7 (2.8–2.7) | 50–2.8 (2.9–2.8) |
| $R_{meas.}$ | 21.5 (152.2) | 9.7 (99.1) | 11.3 (92.0) |
| $I/\sigma I$ | 11.8 (2.5) | 12.0 (1.7) | 11.3 (1.9) |
| Completeness (%) | 99.7 (99.6) | 99.4 (99.2) | 98.6 (89.6) |
| Redundancy | 13.9 (13.7) | 4.2 (4.1) | 4.1 (3.8) |
| $CC_{1/2}$ | 99.8 (82.6) | 99.4 (70.4) | 99.7 (74.4) |
| **Refinement** | | | |
| Resolution (Å) | 50–2.4 | 50–2.7 | 50–2.8 |
| No. reflections | 108,038 | 75,069 | 68,024 |
| $R_{work}/R_{free}$ | 22.9 / 26.9 | 22.1 / 24.6 | 21.9 / 24.4 |
| No. atoms | | | |
| Protein | 14,071 | 14,039 | 14,055 |
| Ligand/ion | 110 | 262 | 275 |
| Water | 110 | 104 | 141 |
| B-factor | | | |
| Protein | 51.6 | 68.0 | 62.5 |
| Ligand/ion | 65.3 | 68.4 | 55.4 |
| Water | 40.8 | 48.8 | 42.8 |
| R.m.s. deviations | | | |
| Bond lengths (Å) | 0.011 | 0.008 | 0.007 |
| Bond angles (°) | 1.38 | 1.17 | 1.13 |
| Atomic coordinates | 6EPG | 6EPH | 6EPI |

**Table 1 Data collection and refinement statistics**

peptidoglycan precursors and most likely does so at a site different from the streptococcal zeta toxins.

**Architecture of the ngε_1/ngζ_1 TA-complex.** In order to gain detailed insights into the zeta toxin encoded by *Neisseria gonorrhoeae*, we aimed at a structural characterization of substrate and product bound states of ngζ_1. To this end, we first determined the crystal structure of the catalytically inactive ngε_1/ngζ_1 (K115A) selenomethionine substituted protein complex at 2.4 Å resolution using single anomalous dispersion experiments (Table 1). We found four heterodimeric ngε_1/ngζ_1(K115A) TA complex molecules located in the asymmetric unit, a complex arrangement which we also observed in solution (Supplementary Fig. 5). Thus, the quaternary structure of the observed heterodimeric ngε_1/ngζ_1 protein complex is substantially different to the hitherto described heterotetrameric oligomeric state for all other epsilon/zeta TA complexes[24,25].

In fact, not only the quaternary structure is set apart from known zeta toxins, but also the tertiary structure and mode of inhibition is very different to other zeta toxins. Strikingly ngε_1 binds as an extended polypeptide stretch onto the molecular surface of ngζ_1 and does not fold into a globular domain (Fig. 3a, b). In contrast, the streptococcal epsilon antitoxins fold into three-helix bundles (Supplementary Fig. 6) and two epsilon molecules form the core of the heterotetrameric epsilon₂/zeta₂ protein complex[24,25]. Furthermore, the C-terminal part of the first helix, the second helix and the connecting loop region of

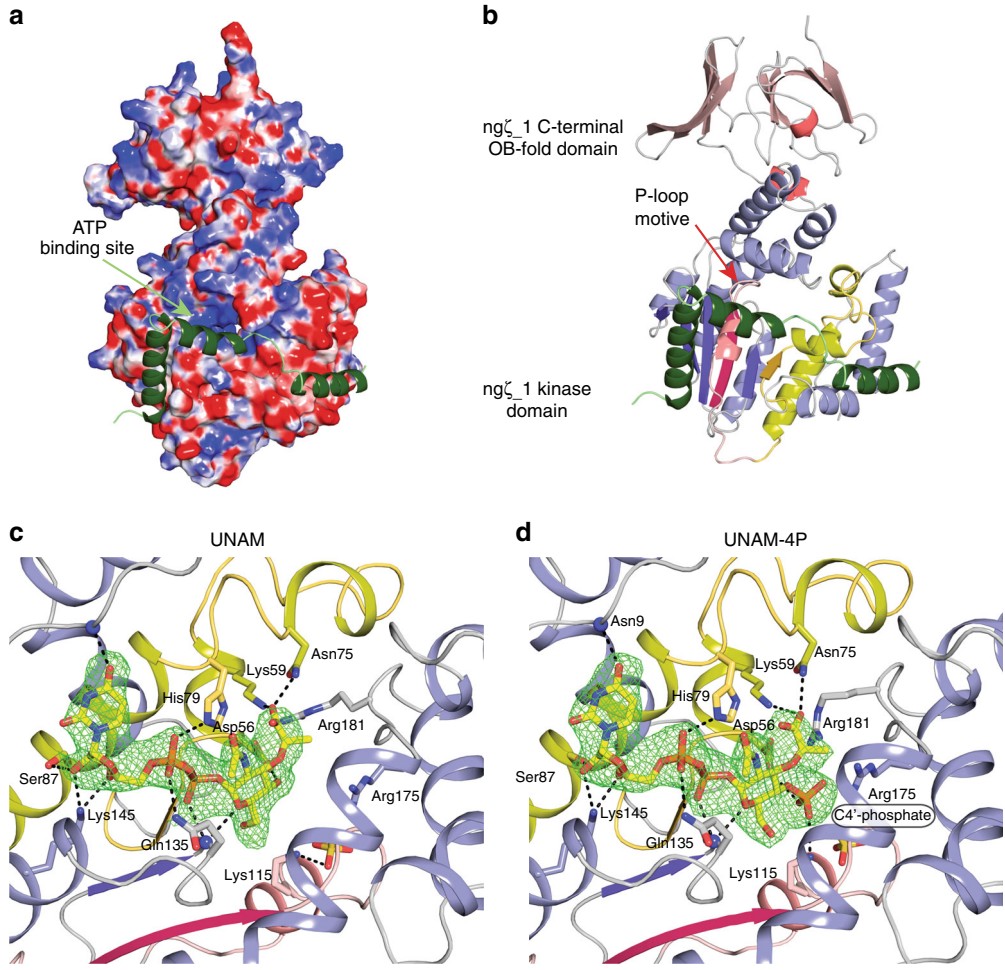

**Fig. 3** Structure of the ngε_1/ngζ_1 TA complex and substrate binding mode. **a** Surface representation of ngζ_1 colored according to the electrostatic surface-potential and ngε_1 depicted as ribbon model (forest-green). **b** Ribbon representation of ngζ_1 at the same orientation as in **a**. Structural elements involved in topological rewiring are highlighted in red (strand)/pink (helix) and dark-yellow (strand)/yellow (helix). The C-terminal OB-fold domain is colored in red. Coloring is performed similar as in Supplementary Fig. 1. **c**, **d** Close-up view of the active site of ngζ_1 with the bound UNAM (**c**) and UNAM-4P (**d**) molecules shown as stick model. Fo-Fc electron density map contoured at 3σ with the ligand omitted before refinement is shown as a mesh

ngε_1 cover the potential ATP binding site (Fig. 3a). In all other epsilon toxins, residues located in the N-terminal region of the first helix of the three-helix bundle protrude into the active site and block ATP binding to the toxic kinase domain[24,25].

In addition, also the tertiary structure of ngζ_1 is different to known zeta toxins, as it consists of two domains (Fig. 3b). The N-terminal domain has the classical fold of a nucleotide kinase to which a DALI search[39] identified zeta toxins[24,25] and AvrRxo1 from *Xanthomonas*[28] to be the closest structural homologues. Although our initial analysis of the primary sequence did not identify these homologues, a structural alignment revealed that two sequence blocks were shuffled by swapping position within the polypeptide chain. Yet, two loop regions were topologically rewired in ngζ_1 when compared with other zeta toxins leading to a similar tertiary structure albeit these two blocks occur at different position within the amino acid sequence (Supplementary Fig. 1). Nevertheless, residues that have previously been shown to be important for enzyme catalysis are found to be structurally conserved in ngζ_1. Finally, the C-terminal domain has the fold of an oligonucleotide/oligosaccharide binding domain (OB-domain) and consists of a five-stranded Greek-key β-barrel that is in an open distorted conformation capped by a short α-helix. Although found in all gonococcal zeta toxins, only the zeta homologue of *Eikenella sp.* NML01-A-086 contained a similar domain.

**ngζ_1 phosphorylates UNAM at the 4′-hydroxy group**. Since we found UNAM to be a superb substrate in vitro compared to all other candidates which we identified by small metabolite extracts, we soaked UNAM into crystals and determined the structure of the complex. Indeed we observed a strong electron density for a UNAM molecule positioned in the active site of all four ngζ_1 polypeptide chains (Fig. 3c). To our surprise, Asp56, which most likely deprotonates the substrate before nucleophilic attack on the γ-phosphate group, is in hydrogen bonding distance to the C4′-OH group of the hexose moiety of UNAM, substantiating phosphorylation of this particular group. Indeed, by soaking purified, ngζ_1 phosphorylated UNAM into our crystals we unambiguously showed that the phosphate group on UNAM was attached to the C4′-OH (Fig. 3d) and not to the C3′-OH group atom, different to what has been described for the hitherto characterized zeta UNAG kinases[21]. We observed a number of interactions between ngζ_1 and UNAM (Fig. 3c, d) and nearly all residues that form hydrogen bonds with UNAM are strictly conserved among homologues, but not in authentic UNAG-3′ kinases (Supplementary Fig. 1). Most importantly, Asn75 and Lys59 form hydrogen bonds with the 1-carboxyethyl group of UNAM which could explain the preference of UNAM over UNAG by ngζ_1. Moreover, Arg181 and Arg175 form a positively charged patch in the active site counteracting the negatively

charged phosphate groups. When comparing the configuration of UNAM bound to ngζ_1 with UNAG bound to *S. pyogenes* zeta toxin[21], we found a UDP-sugar binding mode to the active site that follows a rigid-body rotation by 180° (Supplementary Fig. 7), causing that the C4′-OH and not the C3′-OH group is located in close proximity to the catalytic important aspartic acid (Asp56 in ngζ_1 and Asp67 in *S. pyogenes* zeta toxin). To exclude that UNAG could be phosphorylated at a different site than UNAM we verified the specificity of ngζ_1 towards the less preferred UNAG and confirmed phosphorylation of the C4′-OH group of the *N*-acetylglucosamine moiety by nuclear magnetic resonance (NMR) (Supplementary Fig. 8). In fact, no evidence accumulated from our NMR experiments that indicated any phosphorylation event other than that occurring at the C4′-OH. Similarly, exclusive C3′-OH phosphorylation was shown for streptococcal zeta toxins[21]. Thus, no promiscuity towards the phosphorylation site is given and the observed phosphorylation event is not a side reaction in either case. In conclusion, ngζ_1 performs a yet undescribed phosphorylation of the C4′-OH groups of UDP-sugars.

## A non-discriminative blockade drains peptidoglycan synthesis.
Our small metabolite extracts demonstrated no significant difference in the concentration of the three accumulating, phosphorylated UDP-sugars (Fig. 2a). Yet, our initial qualitative in vitro activity assays pointed to an increased activity of ngζ_1 for UNAM (Supplementary Fig. 4a). Thus, we wondered which factors determine to which extent each individual ngζ_1 product accumulates in vivo. To this end, we first investigated substrate specificity of ngζ_1 and characterized the Michaelis-Menten kinetics for the different identified substrates (Table 2; Supplementary Fig. 9). As expected from our initial, qualitative in vitro characterization, these measurements confirmed the apparent $K_M$ for UNAM to be more than 10-fold smaller than for UNAG or UDP-glucose. Similarly, the determined apparent $k_{cat}$ for UNAM was more than 100-fold higher than for UNAG and UDP-glucose, respectively. This preference became even more prominent when comparing the catalytic efficiency. While this value is rather similar for UNAG and UDP-glucose, it is increased by a factor of approximately 1000 for UNAM, suggesting that UNAM-4P should be the predominant phosphorylated species also in vivo. Thus, we wondered whether UNAG-4P inhibits MurA activity similar as described for UNAG-3P[21] thereby preventing UNAM synthesis by MurA and MurB. This would lead to accumulating cellular levels of UNAG that can be phosphorylated by ngζ_1. Hence, we tested any potential inhibition of MurA by UNAG-4P. However, the reaction kinetics of UNAG phosphorylation by MurA remained unaffected when UNAG-4P was added in equimolar amounts to the natural substrate (Fig. 4a). UNAG-4P is rather a dead-end product and cannot be used by MurA anymore. Similarly, MurB activity was also not affected by the presence of any EP-UNAG-4P and it was not reduced to UNAM-4P by the enzyme (Fig. 4a). Importantly, also MurC activity remained the same and the enzyme could not use UNAM-4P as

potential substrate (Fig. 4a). Apparently, none of these enzymes can use their phosphorylated substrates anymore and thus cannot channel any ngζ_1-phosphorylated peptidoglycan precursor

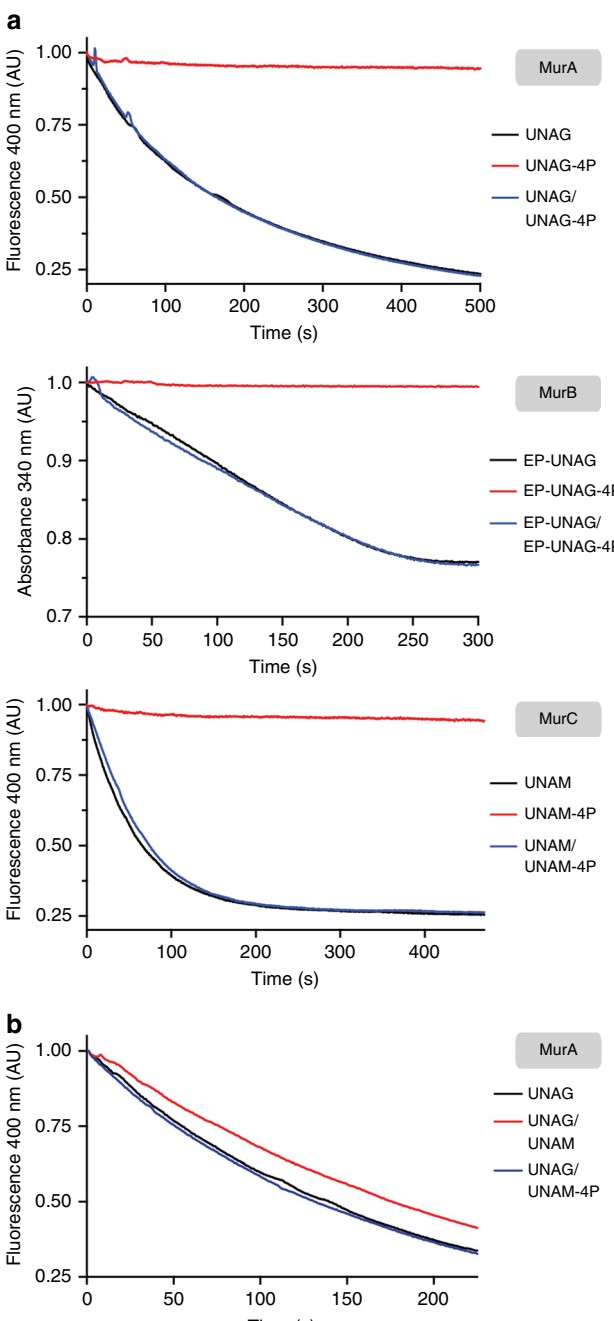

**Fig. 4** Effect of ngζ_1 products on enzymes involved in early peptidoglycan synthesis. **a** Effect of in vitro phosphorylated ngζ_1 products on the activity of the first three enzymes of the peptidoglycan synthesis, MurA (monitoring phosphate release), MurB (monitoring NADPH oxidation) and MurC (monitoring phosphate release). In vitro reactions were perfomed in presence of either the native non-phosphorylated substrate (black; 50 µM UNAG, 30 µM EP-UNAG or 50 µM UNAM), the phosphorylated form (red; 50 µM UNAG-4P, 30 µM EP-UNAG-4P, 50 µM UNAM-4P), or both at equimolar concentrations (blue). Reactions were started upon addition of either 0.25 µM MurA, 50 nM MurB, or 2 µM MurC (time 0 s). **b** Effect of UNAM and UNAM-4P on MurA activity. MurA activity assay using 100 µM UNAG (black) performed in the presence and absence of additional 100 µM UNAM-4P (blue) or UNAM (red)

| Table 2 Michaelis-Menten parameters of ngζ_1 activity | | | |
|---|---|---|---|
| | $K_M$ (mM) | $k_{cat}$ (s$^{-1}$) | $k_{cat}/K_M$ (mM$^{-1}$ s$^{-1}$) |
| UNAM$_{(4\ mM\ ATP)}$ | 0.23 ± 0.02 | 200 ± 7 | 880 |
| UNAG$_{(4\ mM\ ATP)}$ | 2.8 ± 0.3 | 1.72 ± 0.05 | 0.6 |
| UDP-glucose$_{(4\ mM\ ATP)}$ | 6.4 ± 0.7 | 0.71 ± 0.03 | 0.1 |
| ATP$_{(1\ mM\ UNAM)}$ | 2.6 ± 0.4 | 280 ± 16 | 110 |
| ATP$_{(20\ mM\ UNAG)}$ | 0.34 ± 0.02 | 1.76 ± 0.02 | 5.2 |
| ATP$_{(20\ mM\ UDP-glucose)}$ | 0.31 ± 0.05 | 0.51 ± 0.02 | 1.7 |

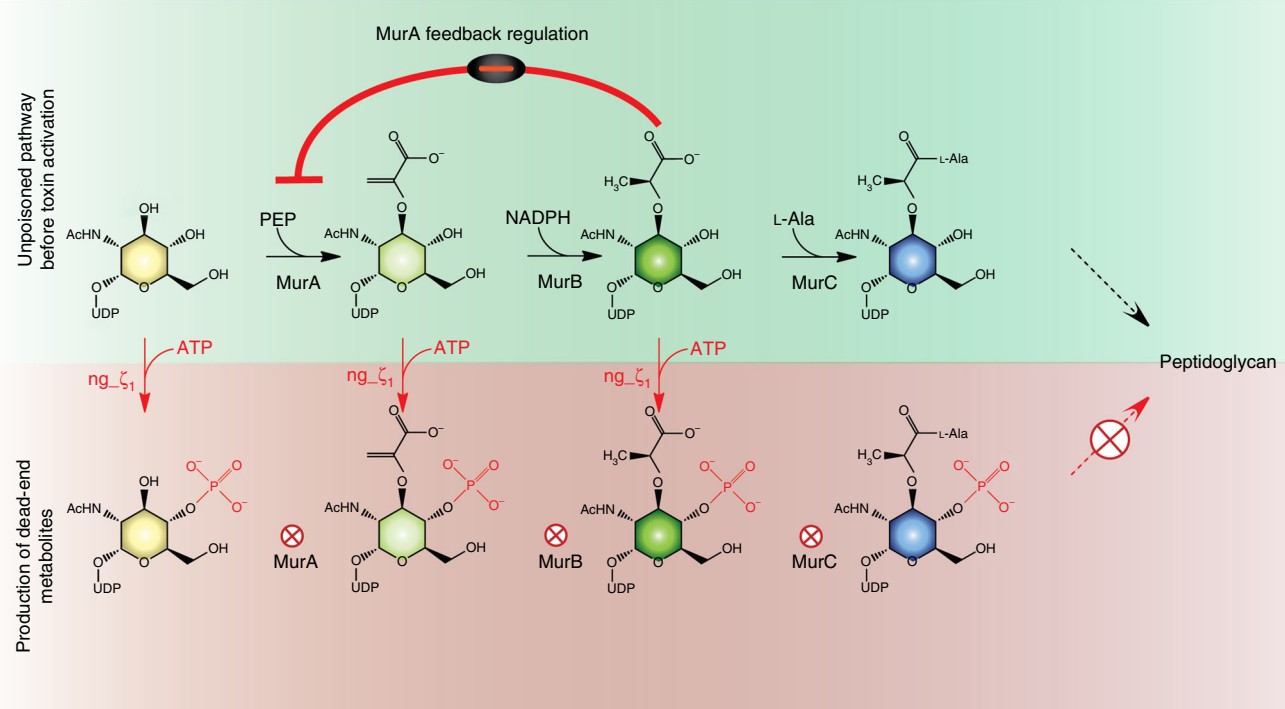

**Fig. 5** ngζ_1 drains precursors from peptidoglycan synthesis at multiple stages. Under normal conditions cytosolic levels of UNAM regulate peptidoglycan synthesis by a negative feedback loop inhibiting MurA. In contrast, once ngζ_1 becomes active, MurA, MurB, and MurC are depleted from their substrates. All phosphorylated precursors are dead-end metabolites; however, none of them seems to directly inhibit any enzyme of early peptidoglycan synthesis

downstream into the pathway. Ultimately, we also tested whether UNAM-4P can inhibit MurA and thus would still mimic the generic feedback loop in early peptidoglycan synthesis[40]. However, this seems also not to be the case as we could not observe such an inhibition by UNAM-4P (Fig. 4b).

In conclusion, our findings strongly suggest that the cellular substrate concentrations are the key determinants for ngζ_1 activity. As proof of principle, we tested EP-UNAG, one of the least abundant precursors in cytosolic peptidoglycan synthesis, and observed robust phosphorylation in vitro already after 30 min of incubation (Supplementary Fig. 10). Yet, we did not observe EP-UNAG-4P in our small metabolite extracts (Fig. 2a) although the resolution of the HPLC-separation would allow for distinguishing any potentially formed EP-UNAG-4P from all other accumulating species (Supplementary Fig. 11). In conclusion these results strongly suggest that the concentration of the accumulating species is dictated by the cytosolic levels and the toxic phenotype of ngζ_1 is due to draining all enzymes of early peptidoglycan synthesis from their substrates (Fig. 5).

## Discussion

We have studied the gonococcal ngζ_1 as the paradigm for a hitherto unknown class of abundant zeta kinases that are wired topologically different when compared to previously characterized zeta toxins of Gram-positive bacteria. Thereby, we identified a yet undescribed enzymatic activity. This proteobacterial zeta toxin phosphorylates UDP-activated sugar species at the C4′-OH group of the hexose moiety which eventually causes a lytic phenotype. The toxic activity is inhibited by co-expression of ngε_1 antitoxin, corroborating that the *nge_1 / ngζ_1* locus encodes for a toxin/antitoxin pair. As expected for a bona-fide type II TA system, we demonstrated protein-protein complex formation between ngε_1 and ngζ_1, which assemble into a heterodimeric complex.

In contrast to previously studied epsilon antitoxins, the ngε_1 wraps around ngζ_1 in an extended conformation and does not protrude into the ATP binding site. In this respect, ngε_1 is more similar to the majority of antitoxins of type II TA systems, such as ParD, RelB, MazE and HigA. For these antitoxins an unfolded structure of the unbound protein was either shown or speculated to cause increased proteolytic instability than when complexed with the cognate toxin[41]. This intrinsic instability is thought to be important in the regulation of toxin activity in the cell[41]. Furthermore, the three-dimensional structure revealed a topological rewiring that seems to distinguish the C4′-OH phosphorylating zeta toxin from the streptococcal C3′-OH phosphorylating zeta subfamily.

Similar to its streptococcal counterparts[21], ngζ_1 was shown to phosphorylate peptidoglycan precursors in vitro and in vivo and cell lysis is a consequence of impairment of peptidoglycan synthesis. However, ngζ_1 uses a broader substrate spectrum and phosphorylates UNAG as well as UNAM and UDP-glucose at their C4′-OH group. Michaelis-Menten kinetics for the different substrates reveal that ngζ_1 has highest catalytic efficiency for UNAM. This leads to a significant accumulation of UNAM-4P in the cell, a dead-end metabolite that cannot be modified by MurC. However, contrasting the reduced catalytic efficiency for UNAG in vitro, considerable amounts of UNAG-4P were detected in vivo. As it is the case for UNAM-4P, also UNAG-4P accumulates as UNAG is also turned into an unsuitable substrate for MurA. In conclusion, both dead-end products of ngζ_1, UNAG-4P and UNAM-4P, drain MurA and MurC from their substrates (Fig. 5).

We showed that the cellular accumulation of UNAG-4P and UNAM-4P is not due to any inhibition of the Mur enzyme cascade but rather is determined by cellular concentrations of the individual precursors. While UNAM is present at an intracellular concentration of only 30–60 μM in *E. coli*[42], the concentration of UNAG is approximately ten-fold higher (100–500 μM)[43–45].

Therefore UNAG-4P accumulated in vivo although it is only slowly converted by ngζ_1. In fact, EP-UNAG, which we could not detect in vivo as accumulating phosphorylated species but is phosphorylated in vitro by ngζ_1, is less abundant by a factor of 100 compared to UNAG[43]. Similarly, in vivo accumulation of the even less preferred UDP-glucose is most likely due to its approximately fifty-fold higher concentration (1.5–1.8 mM)[46]. Altogether, formation of these phosphorylated species depletes bacteria from precursors required for de novo synthesis of cell wall components (Fig. 5).

Although our data imply that the drain of the early steps in peptidoglycan synthesis catalysed by MurA, MurB and MurC is very effective, it seems plausible that the addition of a phosphate group to the C4'–OH group will inhibit further downstream processes as well. A non-redundant cascade of enzymes, namely MurC to MurF, subsequently adds an amino acid stem to UNAM, resulting in UNAM-pentapeptide[27]. MraY anchors the UNAM-pentapeptide to cell membrane forming undecaprenyl-pyrophosphoryl-MurNAc-pentapeptide, known as lipid I. Finally, lipid I is condensed with the C1'–OH group of UNAG via a ß-(1-4)-glycosidic bond by MurG, producing lipid II. This terminates the cytosolic steps of peptidoglycan synthesis as lipid II is flipped either into the periplasmic or extracellular space where transglycosylases catalyze the formation of alternating glycan chain strands. Together with peptide bridges this results in the mesh-like structure of the peptidoglycan layer, which confers rigidity to the cell envelope. Additional inhibitory events are for instance likely at the stage of lipid II formation. MurG which forms the glycosidic bond between a UNAG molecule and lipid I to generate lipid II is highly stereo selective and was shown to exclude the epimer UDP-N-acetylgalactosamine, which just differs in the stereochemistry of the C4'–OH[47]. Accordingly, UNAG-4P will most likely also not be accepted by MurG and ngζ_1 therefore will also impair lipid II formation. Although this requires experimental validation, any phosphorylated lipid II would in turn terminate periplasmic glycan strand synthesis by blocking 1–4 glycosidic bond formation of transglycosylases.

Although our study focused on the effect of ngζ_1 on peptidoglycan synthesis, formation of UNAG-4P but also that of UDP-glucose-4P will additionally affect the synthesis of other cell wall components such as lipopolysaccharides (LPS), colanic acids, teichoic acids and the capsule. Similar to its unique role in peptidoglycan synthesis, UNAG is also the central precursor for lipid A (also known as endotoxin) synthesis, the membrane anchor of LPS of Gram-negative bacteria[48]. The highly conserved UDP-N-acetylglucosamine acyltransferase LpxA, that commits the first step of lipid A synthesis, binds UNAG in a narrow active site[49], and thus likely will not accept UNAG-4P as substrate, similar as we demonstrated for MurA. While lipid A is highly conserved in Gram-negative bacteria, the composition of the residual LPS is much more diverse and strain specific[50]. Yet, D-glucose is a universally incorporated in many of the latter parts of LPS and thus it is tempting to speculate that formation of UDP-glucose-4P might interfere with their synthesis.

Apparently, zeta toxins encoded by Gram-negative and Gram-positive bacteria do not group into a single uniform kinase family but form at least two functionally and structurally differentiated subfamilies. Although similar in their active site architectures, these toxic enzymes vary in their selectivity and specificity for UDP-activated sugar species, but also in the mechanism by which they interfere with vital cellular processes. On one hand, streptococcal toxins stall the first committed step of peptidoglycan synthesis by inhibition, as formation of UNAG-3P by PezT leads to the accumulation of a strong inhibitor of MurA[21]. When a certain level of UNAG-3P is reached the bacterial peptidoglycan synthesis will halt, which can only be relieved by the breakdown of the inhibitor UNAG-3P. On the other hand, ngζ_1 depletes peptidoglycan and LPS from substrates molecules by forming non-inhibiting, dead-end metabolites and interferes at multiple stages in the pathway. Such a depletion mechanism is in constant competition with the de novo synthesis of cell wall components. This potentially allows a tuning of the peptidoglycan synthesis rates, which gradually decreases with an increase of ngζ_1 activity. It seems plausible, that the different mechanisms reflect an adaptation of the zeta toxins to the peptidoglycan and cell wall structure of bacteria. Whereas the thin periplasmic peptidoglycan mesh in Gram-negative bacteria is very sensitive to changes in its synthesis, the thick outer-membrane peptidoglycan of Gram-positive bacteria may tolerate or even require stronger interference. Zeta toxins might differ in their rates and mechanisms of peptidoglycan synthesis inhibition to still allow recovery once the antitoxin is replenished within a certain frame before cell lysis occurs.

Although experimental evidence has not accumulated for Gram-negative zeta toxins so far, their Gram-positive homologues seem to play a role in bacterial pathogenicity as they have been connected with increased virulence[51], reduced sensitivity to antibiotics and competence[35,52], and stable inheritance of mobile genetic elements[53–55]. It seems plausible that ngζ_1 in Gram-negative bacteria will also have an impact on pathogenicity. Ngζ_1 impairs the synthesis of two essential cell wall components, peptidoglycan and LPS, which are common virulence factors of Gram-negative bacteria. LPS and fragments thereof such as lipid A as well as fragments of peptidoglycan are established immunoactive molecules and the innate immune system of any potential host is specialized in recognizing them. Noteworthy, N. gonorrhoea releases a large amount of peptidoglycan fragments such as the Tracheal cytotoxin into the environment[56], which were shown to be involved in pathogenicity[57] and by stalling de novo synthesis of peptidoglycan, ngζ_1 will have an impact on that. On the other side, by interfering with lipopolysaccharide synthesis, ngζ_1 potentially may also modulate endotoxin release of pathogenic Gram-negative bacteria and may also be involved in phase variation of the lipopolysaccharide composition to evade the host immune system[58].

In conclusion, this study showed how a Gram-negative epsilon zeta TA system modifies essential cell wall components and thereby interferes with vital cellular processes. Doubtless, the ngε_1 / ngζ_1 locus encodes for a plasmid-encoded functional TA system that per se has the potential to support stable plasmid maintenance. However, given that ngζ_1 toxicity interferes with the synthesis of pathogenicity-related cellular compounds at multiple levels, it is tempting to speculate that in addition to its influence on genome plasticity, the ngε_1/ngζ_1 might modulate Gram-negative bacteria's pathogenicity; a hypothesis which urges for further detailed studies.

## Methods

**Cloning and protein expression for purification.** Cloning of the individual expression constructs was performed according to standard procedures. The sequence of each individual construct was verified by DNA sequencing. A detailed description for the cloning procedure is given in the Supplementary Methods section and in Supplementary Table 2.

For protein purification, individual constructs were transformed into E. coli strain BL21(DE3)-RIL cells and bacterial cultures were grown in LB-medium with the appropriate antibiotics to an $OD_{600}$ of 0.5 at 37 °C. Subsequently, the cell cultures were transferred to 20 °C and protein expression was induced by addition of 0.5 mM isopropyl-β-D-thiogalactopyranoside. For production of selenomethionine labeled protein, pET28b_ngε_1/ngζ_1(K115A) transformed cells were grown in an amino acid supplemented minimal medium[59]. Cells were harvested by centrifugation and cell pellets were suspended in their appropriate Ni-NTA equilibration buffers (see Supplementary information). Cell walls were broken by sonication and the supernatant was cleared by centrifugation before loading onto the first column. All proteins used in this study were purified by standard chromatographic procedures which are described in detail in the

Supplementary Information. Protein homogeneity was monitored during all steps by Coomasssie stained SDS-PAGE. Protein aliquots were flash-frozen and stored at −80 °C.

**Phenotype characterization of ngε_1/ngζ_1.** Cell lysis accompanied by breakdown of the osmotic barrier was monitored by measuring cell growth at $OD_{600}$ and fluorescence upon propidium iodide (Sigma-Aldrich, St. Louis, USA) influx into ngζ_1 poisoned *E. coli* C41(DE3) cells which are commonly known to tolerate toxin expression much better than BL21(DE3)-RIL. Yet, cells needed to be co-transformed with pET28b_ngε_1 and either pBAD_ngζ_1 or pBAD_ngζ_1 (K115A), respectively, for stable maintenance of toxic ngζ_1 encoding plasmids. However, promotor leakage was sufficient for stable maintenance during all experiments and ngε_1 remained uninduced during all experiments. Overnight cultures were inoculated in LB-medium containing ampicillin and 0.3% (*w/v*) glucose from single colonies and grown at 37 °C. After 1:10 dilution, cell growth was resumed to an $OD_{600}$ of 0.2. Subsequently, this cell culture was diluted with an equal volume of LB medium supplemented with 0.1 mg/ml propidium iodide and 0.2% (*w/v*) L-(+)-arabinose for ngζ_1 or ngζ_1(K115A) expression. Two hundred microliter of these mixtures were transferred into a 96-well plate (Corning 3651, Corning Inc, Corning, USA) and $OD_{600}$ and propidium iodide fluorescence (excitation: 520 nm, emission: 620 nm) was monitored over 90 min using a Varioscan Flash Multimode Reader (Thermo Fisher Scientific, Waltham, USA). Cell cultures for phase contrast and fluorescence microscopy experiments were prepared following the same protocol. Live/dead staining was performed using the LIVE/DEAD BacLight bacterial viability kit (Molecular Probes, Carlsbad, USA).

**Analysis of low-molecular-weight metabolite extracts.** Low-molecular-weight-metabolite extraction and analysis was performed similarly as previously described[21]. Briefly, *E. coli* C41(DE3) cells harboring pET28b(ngε_1) and either pBAD_ngζ_1 or pBAD_ngζ_1(K115A) were grown in 500 ml of LB-medium supplemented with ampicillin to an $OD_{600}$ of 0.3, protein expression was induced upon addition of 0.2% (*w/v*) L-(+)-arabinose and cells were harvested by centrifugation 20 min post-induction. The cell pellets were resuspended in 20 ml ice-cold 80% (*v/v*) aqueous acetonitrile and incubated for 30 min on ice with regular agitation. The supernatant was cleared by centrifugation and the solvent was evaporated. Aliquots were diluted with deionized water, cleared by filtration and adjusted to a common $A_{260}$ of 20 AU. Thirty microliter of sample was applied to a Partisil-5 SAX RACII column (Whatman plc, Maidstone, UK) equilibrated with 5 mM $KH_2PO_4$. Bound metabolites were separated in a linear gradient to 500 mM $KH_2PO_4$. Fractions containing accumulating UDP-sugars were pooled, bound to a MonoQ 5/50 GL column equilibrated with deionized water and eluted in a linear gradient to 1 M ammonium acetate pH 8.0 (32 column volumes (CV)). Solvent and volatile compounds were removed by repeated evaporation and purified compounds were dissolved in deionized water, analyzed with a Bruker maXis II mass spectrometer (Bruker Corporation, Billerica, USA) and fragmented by collision-induced dissociation.

**Oligomeric state determination of the ngε_1/ngζ_1 complex.** The oligomeric state of the ngε_1/ngζ_1 protein complex was determined by multi-angle light scattering (Wyatt, Santa Barbara, USA) coupled to size-exclusion chromatography using a Superdex200 10/300 GL column (GE Healthcare, 50 mM MES-NaOH pH 6.0 and 200 mM NaCl). Forty microliter of either ngε_1/ngζ_1 protein complex or protein standard mixture (Bio-Rad, Hercules, USA) were injected and the recorded data were analyzed using the ASTRA software (Wyatt).

**Synthesis and purification of UDP-sugars.** UNAM was prepared through successive enzymatic conversion of UNAG by MurA and MurB. To avoid potential inhibition of MurA by UNAM[40], 300 μM UNAG was first converted to EP-UNAG by MurA. Therefore, 1 μM MurA was mixed with 1 mM phosphoenolpyruvate (PEP) and 400 μM NADPH in a buffer composed of 50 mM Tris-HCl, pH 8.0, 50 mM NaCl and 10 mM KCl. After 3 h incubation at room temperature, 0.5 μM MurB were added and the reaction was further incubated overnight. The reaction mixture was diluted in deionized water (1:2), loaded onto a MonoQ 10/100 GL column equilibrated with deionized water. Bound residual UNAG and NADP were removed by washing the column with 300 mM ammonium acetate pH 8.0 and UNAM was eluted with 350 mM of ammonium acetate pH 8.0. Fractions containing UNAM were pooled and concentrated by evaporation. Finally, UNAM was desalted by size exclusion chromatography using a Superdex75 10/300 GL column equilibrated with deionized water. UNAM containing fractions were pooled and concentrated by evaporation.

Synthesis of EP-UNAG was performed similar to the production of UNAM but neither MurB nor NADPH was included in the reaction mixture. The reaction was incubated for 2.5 h at room temperature and EP-UNAG was purified by anion exchange and size exclusion chromatography as described for UNAM.

For production of UNAM-4P, 300 μM UNAM was incubated with 10 nM ngζ_1 in preparation buffer (50 mM MES-NaOH pH 6.0, 200 mM NaCl, 0.5 mM EDTA, 14 mM KCl and 2 mM PEP) supplemented with 4 mM $MgCl_2$ and 1 mM ATP. For recycling of ATP from ADP, 7.2 U of pyruvate kinase (PK)/10.8 U of lactate dehydrogenase (LDH) were included in the reaction. After incubation at 25 °C for

6 h, the reaction mixture was diluted in deionized water and loaded onto a MonoQ 5/50 GL column equilibrated with deionized water. Bound ATP was eluted by washing the column with 30 mM $MgCl_2$. Subsequently, the column was washed in four steps with water, 1 mM EDTA pH 8.0, water and 170 mM ammonium bicarbonate, pH 8.0. Pure UNAM-4P was eluted with 1 M ammonium bicarbonate, pH 8.0. Concentrating and desalting of UNAM-4P was performed similar as described for UNAM and EP-UNAG.

The enzymatic conversion of 250 μM EP-UNAG to EP-UNAG-4P by ngζ_1 (1 μM) was performed in preparation buffer supplemented with 1 mM $MgCl_2$, 0.5 mM ATP and 0.6 U PK/0.9 U LDH. Phosphorylated EP-UNAG was loaded onto a Mono Q 5/50 GL column equilibrated with water and eluted in a gradient of 60 CV to 1 M ammonium acetate pH 8.0. Fractions containing EP-UNAG-4P were pooled and concentrated by evaporation. For quantitative removal of ammonium acetate, the pellet of EP-UNAG-4P was re-dissolved in pure water and re-concentrated twice. The correct identity for the different UDP-activated sugar species was verified by ESI-MS and the final concentration of was determined spectroscopically ($\varepsilon_{260} = 10.100$ cm$^{-1}$ M$^{-1}$)[60,61].

**Chromatographic ngζ_1 phosphorylation assays.** Substrate phosphorylation by ngζ_1 was analyzed using a chromatographic assay. Two hundred and fifty micromolar EP-UNAG, UDP-glucose or UNAG and 0.5 mM ATP were mixed in reaction buffer B (50 mM MES pH 6.0, 200 mM NaCl, 1 mM $MgCl_2$, 14 mM KCl, 2 mM PEP, 0.6 U PK and 0.9 U LDH) in a total volume of 180 μl. Reactions were started by addition of 1 μM ngζ_1 and incubated at 25 °C for different time points (5 min, 3, 12, and 24 h). A reaction lacking ngζ_1 was used as a control. The progression of the reaction was followed via anion exchange chromatography. The mixture was applied to MonoQ 5/50 GL column equilibrated with deionized water. After an initial isocratic elution with 200 mM ammonium acetate pH 8.0, a linear gradient to 1 M ammonium acetate was performed (33 CV for UDP-glucose and UNAG, 60 CV for EP-UNAG). The specific retention volume and characteristic 260:280 ratio were used to identify the nucleotide species.

**NMR characterization of UNAG-4P.** UNAG-4P was dissolved in 99.9% $D_2O$ at a concentration of 6 mg/ml and spectra were recorded at 25 °C using a Varian 500 NMR system spectrometer (Agilent). NMR experiments were processed and analyzed using the MestReNova 10.0 (MestreLab Research) software. Atom names and assignment is given in Supplementary Fig. 5.

Signal assignment of $^1H$ NMR experiments (500 MHz, $^1H$-$^1H$-COSY, $D_2O$): δ = 7.98 (d, 1H, $^3J_{HH}$ = 8.2 Hz, H-U6), 6.0–5.98 (m, 2H, H-U5, H-1′), 5.53 (dd, 1H, $^3J_{HH}$ = 7.3 Hz, $^3J_{HP}$ = 3.2 Hz, H-1), 4.40–4.37 (m, 2H, H-2′, H-3′), 4.30 (m, 1H, H-4′), 4.28–4.25 (m, 1H, H-5′a), 4.22–4.18 (m, 1H, H-5′b), 4.08–3.97 (m, 4H, H-2, H-3, H-4, H-5), 3.90–3.84 (m, 2H, H-6a, H-6b), 2.09 (s, 3H, H-8) ppm.

Assignment of $^{13}C$ NMR experiments (152.7 MHz, APT, HSQC, $D_2O$): δ = 174.63 (C, C=O), 166.14 (C, C=O), 151.72 (C, C=O), 141.55 (CH, C-U6), 102.59 (CH, C-U5), 94.03 (CH, $J_{CP}$ = 6.2 Hz, C-1), 88.38 (CH, C-1′), 83.16 (CH, $J_{CP}$ = 9.0 Hz, C-4′), 73.71 (CH, C-3′), 73.25 (CH, $J_{CP}$ = 4.7 Hz, C-5), 72.03 (CH, $J_{CP}$ = 5.7 Hz, C-4), 70.41 (CH, C-2′), 69.57 (CH, C-3), 64.90 (CH$_2$, $J_{CP}$ = 6.2 Hz, C-5′), 60.11 (CH$_2$, C-6), 53.35 (CH, $J_{CP}$ = 8.5 Hz, C-2), 22.00 (CH$_3$, C-8) ppm.

Assignment of $^{31}P$ NMR decoupling experiments (202.4 MHz, $D_2O$): δ = 0.91 (s, P-4), −11.46 (d, $J_{PP}$ = 19.7 Hz, P-5′), −13.16 (d, $J_{PP}$ = 20.0 Hz, P-1) ppm.

**Structure determination.** Crystals of wild type ngε_1/ngζ_1 or selenomethionine labelled ngε_1/ngζ_1(K115A) protein complex were grown at a concentration of 22 mg/ml using a sitting drop vapor diffusion setup. Best crystals grew as stacked plates using a reservoir solution of 100 mM sodium citrate pH 4.5 and 1.8 M $(NH_4)_2SO_4$. Crystals were transferred to a solution containing 100 mM sodium citrate pH 4.5, 2 M $(NH_4)_2SO_4$ and 20 % (*v/v*) propylene glycol and flash cooled in liquid nitrogen. UNAM or UNAM-4P at a concentration of 20 mM were added to the cryo-protectant and incubated for 10 min during soaking experiments.

Se-SAD and native diffraction data were collected at the Swiss Light Source X10SA beamline and processed with the XDS software package[62], 5% of the reflection were randomly selected for calculation of $R_{free}$ and inherited to all datasets. Initial phases were obtained using the SHELXC and SHELXD[63] from Se-SAD on ngε_1/ngζ_1(K115A) protein complex crystals. Phases from the Se-substructure obtained from SHELXD were improved using Phaser[64], Parrot[65] and DM[66]. An initial model was built using Coot[67] and refined using the simulated annealing protocol of CNS[68]. Subsequently, the model was improved in iterative cycles of manual building in Coot and refinement using Refmac[69]. Final structure validation was performed using MolProbity[70]. Figures were prepared using PYMOL[71].

**ngζ_1 kinase activity assays.** Phosphorylation of different UDP-activated sugar species by ngζ_1 was monitored using a coupled spectroscopic assay in which formation of ADP is coupled to consumption of NADH[72]. Conversion of NADH to NAD$^+$ was monitored spectroscopically at 340 nm (Jasco V650, Jasco, Mary's Court, USA). All reactions were performed in assay buffer (50 mM MES-NaOH pH 6.0, 200 mM NaCl, 1 mM EDTA, 6 mM $MgCl_2$, 14 mM KCl, 1 mM PEP, 0.25 mM NADH, 1 mg/ml BSA, 0.6 U PK and 0.9 U LDH) in a total volume of 100 μl, unless otherwise explicitly stated.

In qualitative experiments, where the apparent velocities of different UDP-activated sugars were compared, 10 nM of ngζ_1 and 4 mM $Mg^{2+}$-ATP were added to the assay buffer and incubated until a stable baseline was observed. The reaction was started upon addition of either 50 μM of UNAG, UNAM, or UDP-glucose.

For the titration experiments aiming at determination of the Michaelis Menten parameters for UDP-sugar species, 4 mM $Mg^{2+}$-ATP and 200 nM (in case of UNAG and UDP-glucose) or 5 nM (in case of UNAM) ngζ_1 were added to the assay buffer. Reactions were started by addition of varying concentrations of different substrates (UNAG: 0.31–40 mM, UDP-glucose: 0.31–20 mM, UNAM: 23–750 μM). For UNAM titration the amount of PK and LDH was increased threefold. While a stable baseline was observed when using a low enzyme concentration (5 nM ngζ_1), at 200 nM ngζ_1 a weak basal ATPase background activity was observed. This apparent basal reaction velocity was determined to be in the range of 0.03–0.13 $s^{-1}$ and was subtracted for baseline correction in these experiments. In experiments aiming at determination of the Michaelis Menten parameters for ATP, 200 nM (in case of UNAG or UDP-glucose) or 10 nM (in case of UNAM) ngζ_1 and varying concentrations of $Mg^{2+}$-ATP from 31.25 μM–4 mM were added to the reaction mixture and incubated until a stable baseline was reached. In case of a weak basal ATPase activity a baseline correction was performed as described above. The reaction was started by addition of either 20 mM UNAG or UDP-glucose or 1 mM UNAM.

The apparent initial velocities for each measurement were determined by the Spectra Manager Analysis software (JASCO), converted to apparent $k_{cat}$ and each measurement was fitted to the Michaelis Menten equation with GraphPad Prism v.5 (GraphPad, La Jolla, USA).

**Activity of peptidoglycan synthesis enzymes**. MurA and MurC activities were monitored in an assay that couples the release of inorganic phosphate to the cleavage of fluorescent 7-methylguanosine[73] by purine nucleoside phosphorylase (PNPase, Sigma-Aldrich, St. Louis, USA). The resulting decrease in fluorescence at 400 nm (bandwidth 5 nm) was measured with a spectrofluorometer (FP-8500, JASCO) using 300 nm as the excitation wavelength (bandwidth 5 nm). MurA reactions were performed in a total volume of 400 μl in a buffer containing 50 mM Hepes-NaOH pH 7.5, 50 mM NaCl, 1 mM PEP, 0.5 mM EDTA, 50 μM 7-methylguanosine and 0.3 U PNPase supplemented with either 50 μM UNAG, 50 μM UNAG-4P or both (50 μM each). After a stable baseline was reached the reaction was started upon addition of 0.25 μM MurA. A potential inhibitory effect of UNAM and UNAM-4P on the MurA activity was tested through the addition of 100 μM UNAM or UNAM-4P to a reaction containing 100 μM UNAG as substrate.

Similarly, MurC activity was measured in a total volume of 400 μl in a buffer containing 50 mM Hepes-NaOH pH 7.5, 0.1 mM EDTA, 10 mM $MgCl_2$, 300 μM $Mg^{2+}$-ATP, 0.5 mM L-Ala, 0.3 U PNPase, 50 μM 7-methylguanosine, and either 50 μM UNAM, 50 μM UNAM-4P or both. After reaching a stable baseline 2 μM MurC were added to start the reaction.

MurB activity was followed spectroscopically by directly monitoring the oxidation of NADPH at 340 nm in buffer containing 50 mM Tris-HCl pH 8.0, 50 mM KCl, and 150 μM NADPH in a total volume of 100 μl. Subsequently, either 30 μM EPUNAG, 30 μM EPUNAG-4P or both (30 μM each) were added to the reaction which was started upon addition of 50 nM MurB.

**Data availability**. Coordinates and structure factor amplitudes have been deposited in the Protein Data Bank under accession codes 6EPG, 6EPH and 6EPI. Other data are available from the corresponding author upon reasonable request.

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

## Acknowledgements

We thank C. van der Does (University Freiburg, Germany) for providing plasmid DNA and are grateful to T. Barends, J. Reinstein, and T. Clausen for helpful discussions. We acknowledge C. Roome for support of the crystallographic software and IT, F. Jungblut for technical assistance and the PXII staff for their support in setting up the beamline. Diffraction data were collected at the Swiss Light Source, beamline X10SA, Paul Scherrer Institute, Villigen, Switzerland. We are grateful to I. Schlichting for continuous encouragement and support. This work was financially supported by the Max Planck Society. A. M. is supported by the Chica and Heinz Schaller Foundation, Heidelberg, Germany.

## Author contributions

A.R. and M.P. designed and performed most of the experiments and wrote the manuscript. T.K. performed some of the kinetic experiments and nucleotide activated sugar purifications and wrote the manuscript. R.S. purified and crystallized the protein complex. C.B. performed NMR and analyzed the data. A.M. performed together with A.R. the crystallographic experiments, was involved in experimental design and wrote the manuscript.

## Additional information

**Competing interests:** The authors declare no competing interests.

