## [Peer Review File(PDF 651 kb) · Nature Communications]

Reviewers' comments:

Reviewer #1 (Remarks to the Author):

Overall, this is a very interesting paper reporting on the characterization of a novel toxin of the Zeta family of the Epsilon-Zeta antitoxin-toxin system from *Neisseria gonorrhoeae* which adds considerably to our knowledge of these family of toxins and the biology of bacterial toxin-antitoxin systems. The so-called ng_ζ1 toxin is found in the pEP5259 conjugative plasmid of *N. gonorrhoeae* which contained a second epsilon-zeta homologue, ng_ε2/ng_ζ2. The ng_ζ1 toxin was found to be a kinase that differs considerably to the canonical Zeta toxins from the streptococci (i.e., Zeta from *S. pyogenes* and PezT from *S. pneumoniae*) with a broader substrate specificity and phosphorylates at the C4'-OH group of hexose sugars (as compared to Zeta/PezT which phosphorylates UDP-N-acetylglucosamine, UNAG at the C3'-OH, to form UNAG-3P). The corresponding ng_ε1 antitoxin also differs from the streptococcal Epsilon antitoxins by binding to ng_ζ1 as an extended polypeptide that covers the active ATP-binding site of ng_ζ1 instead of protruding into the ATP-binding site to block ATP from binding to the kinase domain. In this reviewer's opinion, the novelty of these findings warrants publication in *Nature Communications*, subject to the authors making several minor revisions to clarify certain sections of their manuscript.

- In this reviewer's opinion, the title is too generalized. A more specific title, such as "The ng_ζ1 toxin of the gonococcal Epsilon-Zeta toxin/antitoxin homologue drains precursors for cell wall synthesis" is recommended.

- **Abstract:**

- o Line 4: "...genomes of many bacteria", not "...genome of many bacteria"

- o Line 5: "...by inhibiting peptidoglycan synthesis" seems more appropriate than "...by perturbing peptidoglycan synthesis"

- o A sentence to describe the novel structure of the ng_ε1 antitoxin as compared to the streptococcal Epsilon antitoxins seems warranted in the Abstract

- **Introduction:**

- o Page 4, paragraph 1: What is the size of the pEP5259 conjugative plasmid? How closely related are the ng_ε1/ ng_ζ1 and ng_ε2/ ng_ζ2 systems? Are these gonococcal epsilon-zeta systems found in other conjugative plasmids of *N. gonorrhoeae*?

- o Page 4, paragraph 2: "...dead-end metabolites resulting in cell death by draining the precursors for cell wall synthesis" (suggest adding "the precursors for"); "As phosphorylation of the C4'-OH group.." (missing "the")

- o A comment on the EzeT Zeta homologue from *E. coli* that was characterized by the authors [Rocker and Meinhart, *Mol. Microbiol* (2015)] would be helpful in elaborating on the statement that "...these homologues are also highly prevalent in Gram-negative bacteria" (page 4). Likewise, a sentence of two describing the Zeta homologue, AvrRxo1 from *Xanthomonas* from the authors [Schuebel et al., *JBC* (2016)] and others [Triplett et al., *PLoS One* (2016); Shidore et al., *PLoS Pathogens* (2017)] would help in illustrating the diversity of the Zeta family of toxins.

- **Results and Discussion:**

- o Although the focus of the paper is on the ng_ε1/ ng_ζ1 system, this reviewer feels it will be helpful if some basic data on the second Epsilon-Zeta system found in the *N. gonorrhoeae* plasmid, ng_ε2/ng_ζ2 is presented here including (i) the sequence similarities between the two systems (see above comment) – perhaps include ng_ζ2 in the amino acid sequence alignment shown in Supplemental Figure 1, (ii) discussion on whether the findings for the ng_ε1/ng_ζ1 system would be equally valid for the ng_ε2/ng_ζ2 system and what are the implications for having two such systems on the same conjugative plasmid?

- o How extensive are the occurrence of the ng_ε1/ ng_ζ1 subfamily of the Epsilon-Zeta TA system among the sequenced *N. gonorrhoeae* genomes that are available in the database? The first paragraph of the Results (page 5) indicates that these closely related homologues are found in 15 different, mainly pathogenic bacteria. However, Supplementary Figure 1 only showed the sequences for the homologues from *Campylobacter gracilis*, *Burkholderia pseudomallei* and *Sphingobium cloacae*. It would be helpful to include a Supplementary Table listing the bacterial genomes in which these ng_ζ1 homologues are found and their respective accession numbers. Is the ng_ζ1 subfamily constrained to Gram negative bacteria only? For the legend to Supplementary Figure 1, please list the accession numbers/locus IDs for the ng_ζ1 homologues indicated in the multiple sequence alignment.
- o Page 9: it was stated that the oligonucleotide/oligosaccharide-binding domain (OB-domain) is only found in the gonococcal Zeta toxins and the homologue from *Eikenella* sp. (NML01-A-086). It would perhaps be helpful to include this Zeta homologue in the Supplementary Figure 1 alignment.
- o Page 7: it was stated that "Detecting phosphorylated UNAM was quite surprising as the hitherto characterized zeta toxins phosphorylate UNAG at the C3'-OH group which is blocked by a lactoyl group in case of UNAM. It would be useful to show the structures of UNAG and UNAM in a Supplementary Figure to better illustrate this fact.
- o Page 10: it would be helpful to label and indicate the C4'-P group in Figure 3d.
- o Page 13: intracellular concentrations of UNAM and UNAG are stated here and from the references that were cited, these numbers were likely obtained from *E. coli*. Would this be valid for *N. gonorrhoeae* as well?
- o Page 14: "...where transglycosylases catalyze the formation of alternating glycan chain strands" (insert underlined words), as the transglycosylases themselves do not form the alternating glycan chain strands. What are the enzyme(s) that catalyze the formation of peptide bridges which lead to the formation of mesh-like architecture of the peptidoglycan layer?
- o Page 15: "...reflect an adaptation of the zeta toxins to the peptidoglycan and cell wall structure of bacteria" (suggest addition of "and cell wall" as ng_ζ1 may affect the synthesis of the LPS layer of Gram-negative bacteria; "...very sensitive to changes..." (not "...sensitive on changes.."; final line page 15).

- **Material and Methods:**

- o Page 18: it was stated that for protein expression, *E. coli* BL21 (DE3)-RIL was used but subsequent experiments listed the use of *E. coli* C41(DE3) cells. Please provide the rationale for using this strain of *E. coli*.
- o Pages 20-21: synthesis and purification of UNAM, EP-UNAG, UNAM-4P and EPUNAG-4P. These UDP-derived compounds were synthesized by enzymatic conversion of the relevant substrates. Were there any steps taken to validate the purified products? Please provide the relevant citation(s) for the ε₂₆₀ value for the UDP-derived sugar species given in the last line of page 21.

Reviewer #2 (Remarks to the Author):

In this paper, Rocker et al. provide a thorough functional and structural analysis of a novel type of Zeta-Epsilon toxin-antitoxin system. The data included are extensive and all point to this TA system as being both functionally and structurally divergent from previously characterised systems in this group. Altogether, I think the data constitute a solid amount of novel data and thus warrant publication in *Nature Communications*. However, in particular the structural analysis suffers from several short-comings as detailed below, which should be rectified before publication can be considered.

Supplementary Figure 1 and the whole concept of "shuffling" or rearrangement of the protein domains is not clear. I suggest improving the descriptions on p. 6 and 9 as well as redesigning the figure to

make this clearer. I don't understand the grey lines in the figure and there is not much help in the legend.

Please also include an alignment for the antitoxin demonstrating that it is very divergent as claimed.

Page 8. Please include the single chain molecular masses for both toxin and antitoxin to allow the reader to assess the analysis of the complex stoichiometry.

Page 8. I am missing a structural comparison to the heterotetrameric complexes, either in Fig. 3 or as a supplementary figure. I assume that the authors have checked the crystal packing carefully for a possible tetramer in this case?

Table 1. PDB entries missing.

Figure 3a. I suggest you include an overview of the TA complex in cartoon representation as panel A. It is very difficult to understand the fold and mode of antitoxin interaction from the current figure with an electrostatic surface on the toxin part. And perhaps also a zoomed view of the TA interaction? I am also missing a comparison to the previously described, helical bundle antitoxins. Are there any structural similarity at all or are they completely different?

Figure 3c. The legend says that the map is $F_o - F_c$ but it is not clear to me if the ligand was omitted during refinement. In any case, you should show a complete unbiased electron density map calculated before the ligand was included in the refinement. Otherwise there is a slight risk of phase bias. I am also missing a structural comparison to the 3'OH reacting enzymes here to show why this toxin phosphorylates the 4'OH.

Figure 3d. Labels for the sugar atoms (+phosphate) could improve the figure.

Supplementary Figure 4. I don't understand why the chromatogram is printed so small when there is a whole page and this is a supplementary file. Please enlarge so one can see the light scattering data. The raw light scattering data should be shown as dots, not with a line.

Minor issues:

Page 6. This is perhaps pedantic, but I don't think you can question whether ng epsilon 1 is a "bona fide TA system at all" (end of paragraph), you can question whether it's a TA system and show that it's bona fide.

Reviewer #3 (Remarks to the Author):

Toxin - antitoxin (TA) modules belonging to the Epsilon (ϵ) / Zeta (ζ) family have been primarily described in Gram-positive bacteria where they are prevalent, but is also present in some Gram-negative species. The present, very interesting contribution, describes the discovery of an extended target spectrum of the Zeta/ ζ toxin and the structure of the ϵ/ζ TA complex from the Gram-negative gammaproteobacterium *N. gonorrhoea*. The work is solid and the conclusions are justified. Overall, the work constitutes an important contribution to the field that deserves publication in a general journal. However, as described in some detail below, parts of the presentation, including some phrasings should be improved before publication. The most important improvement relates to point 9 as described below.

1. The title should be more focused to better cover the main content of the manuscript. For example: "Zeta toxin of gonococci lyse bacteria by phosphorylation of multiple metabolites involved in cell wall

synthesis" or something like that.

2. Abstract, line 6: "zeta-related" is imprecise. They must mean "epsilon/zeta-homologous". The Zeta toxins exhibit primary, secondary and tertiary similarities and are therefore homologous. By inference, the modules/loci/systems must be homologous and not analogous. In that vein, it would be pertinent to include a supplementary Figure showing the genetic structure of the Gram-negative ϵ/ζ modules and compare it to that of ϵ/ζ modules of Gram positives.

3. P3, line 7: It would be fair here to include references to the discovery of ribosome-dependent and independent TA-encoded toxins/mRNases.

4. P3, line 9: same with reference 17, replace it or include a reference to the discovery of membrane-damaging toxins.

5. P3, line 13 and other place: Replace "genuine" with "canonical" or "bona fide"

6. P4, line 1: unclear what "zeta-dormant" means. Probably the authors means when Zeta toxin is inactive (due to being in complex with Epsilon). Please amend.

7. Supp. Fig. 1: It would be useful if the authors could add a phylogenetic tree of Zeta toxins, indicating the organisms in which the toxins are present.

8. P5, line 10: delete "at all"

9. P8: consider moving Table 1 to Table S1

10. P11: The section starting with "A non-discriminative...." must be presented in a clearer way. One possibility is to add an overview Figure showing that pathway of Cell Wall synthesis, showing intermediates and enzymes, and indicating which metabolite intermediates are phosphorylated by Zeta toxins from Gram-negative and -positive bacteria. Since the manuscript has only 4 main Figures there is ample space for an additional Figure.

11. Consider shortening of the Discussion.

We are grateful to all the positive and encouraging responses from all reviewers. We highly appreciate the very constructive comments, which we have addressed in the current revised version of our manuscript. We feel that the manuscript has now substantially improved and will give our response to their comments in the following section.

Reviewer #1 (Remarks to the Author):

Comment 1: *In this reviewer's opinion, the title is too generalized. A more specific title, such as "The ng_ζ1 toxin of the gonococcal Epsilon-Zeta toxin/antitoxin homologue drains precursors for cell wall synthesis" is recommended.*

We agree that the title was too generic and have changed it to:

"The ng_ζ1 toxin of the gonococcal Epsilon/Zeta System drains Precursors for Cell Wall Synthesis"

(See also reviewer #3 comment 1)

Comment 2: *Abstract: Line 4: "...genomes of many bacteria", not "...genome of many bacteria"*

Done.

Comment 3: *Line 5: "...by inhibiting peptidoglycan synthesis" seems more appropriate than "...by perturbing peptidoglycan synthesis"*

In fact, we did not observe any inhibition of any tested enzymes of peptidoglycan synthesis by the ng_ζ1 toxin's products and are thus hesitant to include the word inhibition in this sentence.

Comment 4: *A sentence to describe the novel structure of the ng_ε1 antitoxin as compared to the streptococcal Epsilon antitoxins seems warranted in the Abstract*

Given the limited number of words, we have included this short statement in the abstract:

...In contrast to the previously studies epsilon/zeta TA systems, ng_ε1 has an epsilon-unrelated fold and ng_ζ1 has broader substrate specificity...

Comment 5: Page 4, paragraph 1: What is the size of the pEP5259 conjugative plasmid? How closely related are the ng_ε1/ng_ζ1 and ng_ε2/ng_ζ2 systems? Are these gonococcal epsilon-zeta systems found in other conjugative plasmids of *N. gonorrhoeae*?

In response to the question about the size of the plasmid:

We thank the reviewer for this comment but also want to apologize, as we found a typographical error in the manuscript. The correct name of the plasmid is pEP5289 (Pachulec & van der Does, PLoS one, 2010) and the size is 42,004 bp. However, since this is the name of a single clonal isolate that was entirely sequenced (Pachulec & van der Does, PLoS one, 2010) and used as template for PCR amplification, we have revised the manuscript accordingly and the plasmid size is now given in the supplemental information. pEP5289 belongs to the 25.2 MDa “Dutch” type plasmid harbouring a *tetM* determinant. According to Pachulec and van der Does, the ng_ε1/ng_ζ1 locus is also found within the genetic load region of other conjugative plasmids such as the conjugative 24.5 MDa as well as the 25.2 MDa “American” type plasmid. We have revised the introduction accordingly and added this information to the introductory part.

In response to the question how closely related the ng_ε1/ng_ζ1 and ng_ε2/ng_ζ2 systems are: As we describe in the manuscript, the gonococcal ng_ε1/ng_ζ1 contains a domain shuffling when compared to streptococcal zeta toxins and thus their primary sequence cannot be simply aligned. In contrast, the gonococcal ng_ε2/ng_ζ2 as well as the ng_ε3/ng_ζ3 are very similar in their amino acid sequence to zeta toxins and share around 40 % sequence similarity with the streptococcal zeta toxin, an information we have included in the manuscript (see also our response comment 8 from reviewer 1). However, we are reluctant to discussing anything else than this sequence homology, as preliminary results showed that the two systems are neither like streptococcal zeta nor to ng_ζ1 toxins (see response to comment 8 from reviewer 1). We would thus like to exclusively concentrate on ng_ζ1 in the current manuscript.

In response to the question asking for the distribution of epsilon/zeta systems in other conjugative plasmids of *N. gonorrhoeae*: This has been discussed in detail in the original paper (Pachulec & van der Does, PLoS one, 2010) and a summary of the latter is now included in the manuscript.

Comment 6: Page 4, paragraph 2: “...dead-end metabolites resulting in cell death by draining the precursors for cell wall synthesis” (suggest adding “the precursors for”); “As phosphorylation of the C4'-OH group..” (missing “the”)

Done and reads now:

...dead-end metabolites resulting in cell death by draining the precursors required for cell wall synthesis. As phosphorylation of the C4'-OH group...

Comment 7: A comment on the EzeT Zeta homologue from *E. coli* that was characterized by the authors [Rocker and Meinhart, *Mol. Microbiol* (2015)] would be helpful in elaborating on the statement that "...these homologues are also highly prevalent in Gram-negative bacteria" (page 4). Likewise, a sentence of two describing the Zeta homologue, AvrRxo1 from *Xanthomonas* from the authors [Schuebel et al., *JBC* (2016)] and others [Triplett et al., *PLoS One* (2016); Shidore et al., *PLoS Pathogens* (2017)] would help in illustrating the diversity of the Zeta family of toxins.

We have included a short paragraph in the introduction as suggested by the reviewer.

Comment 8: Although the focus of the paper is on the ng_ε1/ng_ζ1 system, this reviewer feels it will be helpful if some basic data on the second Epsilon-Zeta system found in the *N. gonorrhoeae* plasmid, ng_ε2/ng_ζ2 is presented here including (i) the sequence similarities between the two systems (see above comment) – perhaps include ng_ζ2 in the amino acid sequence alignment shown in Supplemental Figure 1, (ii) discussion on whether the findings for the ng_ε1/ng_ζ1 system would be equally valid for the ng_ε2/ng_ζ2 system and what are the implications for having two such systems on the same conjugative plasmid?

We fully agree with the reviewer that the existence of two related TA systems on a single plasmid is striking and would be worth of further investigations.

In response to (ii): Indeed, we have preliminary data on the specificity of the ng_ζ2 as well as the ng_ζ3 toxins. Although the latter two toxins are much more related to streptococcal zeta toxins in their amino acid sequence, we found that both phosphorylate UDP-sugars but at a different OH-group than streptococcal zeta or the ng_ζ1 toxins. However, these results are too preliminary to be mentioned or even included in the current manuscript, which exclusively focuses on the functional mechanism of ng_ζ1. Thus, we feel that including ng_ζ2 or ng_ζ3 in any sequence alignment would suggest that they are zeta like UNAG-3' kinases which we consider as highly misleading given our preliminary results.

In response to (i): As discussed in detail in the manuscript, only at the level of a structural alignment, ng_ζ1 turns out to be homologous to streptococcal zeta toxin. However, any useful

amino acid sequence alignment requires artificial manipulation of amino acid sequence blocks due to the domain swap as illustrated in the Supplemental Figure 1. We thus refrained from giving any sequence homology of ng_ζ1 with streptococcal zeta toxins or ng_ζ2 or ng_ζ3 (see also our response to comment 5 of reviewer 1). However, as pointed out by reviewer 2 (comment 1) we realized that our description in the figure legend, which we have revised to make it more comprehensible, was not amenable in its original form.

Comment 9: *How extensive are the occurrence of the ng_ε1/ ng_ζ1 subfamily of the Epsilon-Zeta TA system among the sequenced N. gonorrhoeae genomes that are available in the database? The first paragraph of the Results (page 5) indicates that these closely related homologues are found in 15 different, mainly pathogenic bacteria. However, Supplementary Figure 1 only showed the sequences for the homologues from Campylobacter gracillis, Burkholderia pseudomallei and Sphingobium cloacae. It would be helpful to include a Supplementary Table listing the bacterial genomes in which these ng_ζ1 homologues are found and their respective accession numbers. Is the ng_ζ1 subfamily constrained to Gram negative bacteria only? For the legend to Supplementary Figure 1, please list the accession numbers/locus IDs for the ng_ζ1 homologues indicated in the multiple sequence alignment. A blast search currently reveals 46 entries for gonococcal ng_ζ1 with 100 % amino acid sequence identity, an information that is now included in the manuscript. Furthermore, Supplementary Table 1 has been included listing all different bacterial organisms in which a ng_ζ1 homologue is found. Also accession numbers for the homologues aligned in Supplemental Figure 1 are given.*

Comment 9: *Page 9: it was stated that the oligonucleotide/oligosaccharide-binding domain (OB-domain) is only found in the gonococcal Zeta toxins and the homologue from Eikenella sp. (NML01-A-086). It would perhaps be helpful to include this Zeta homologue in the Supplementary Figure 1 alignment.*

As seen in the new Supplemental Table 1, the homologue from Eikenella sp. (NML01-A-86) has the highest (74 % amino acid sequence identity) similarity to the gonococcal ng_ζ1. Thus, we would heavily bias the sequence alignment towards the conservation of the two homologues thereby losing the representative character we aimed for. Furthermore, we would like to apologize that the caption of Supplemental Fig. 1 was somehow misleading. It did not

explicitly state that an alignment of just the kinase domain has been shown. In fact, other homologues have different extensions or even lack them.

Comment 10: Page 7: it was stated that “Detecting phosphorylated UNAM was quite surprising as the hitherto characterized zeta toxins phosphorylate UNAG at the C3’-OH group which is blocked by a lactoyl group in case of UNAM. It would be useful to show the structures of UNAG and UNAM in a Supplementary Figure to better illustrate this fact.

We appreciate this suggestion and the manuscript now includes Supplemental Fig. 2 showing the structures of UNAG and UNAM.

Comment 11: Page 10: it would be helpful to label and indicate the C4’-P group in Figure 3d.

Excellent idea and done.

Comment 12: Page 13: intracellular concentrations of UNAM and UNAG are stated here and from the references that were cited, these numbers were likely obtained from *E. coli*. Would this be valid for *N. gonorrhoeae* as well?

Correct, this numbers were obtained from *E. coli*. The main purpose of this discussion is based on our findings that our kinetic studies would on a first glance suggest that UNAM-4P would be found to accumulate in excess over UNAG-4P *in vivo*. However, our analysis of small metabolite extracts from poisoned *E. coli* cells (Fig. 2a) does not support this. In fact, as outlined in the manuscript, the cellular concentration of peptidoglycan precursors inherently dictates the observed concentration after ng_ζ1 phosphorylation, unless that none of the enzyme is inhibited by ng_ζ1’s products. In fact, the latter we unambiguously show. Thus, comparing them with numbers from *N. gonorrhoeae* would be rather incorrect at this place. Nevertheless, independent what the cytosolic ratio of UNAG and UNAM would be and how they would accumulate in their ng_ζ1 phosphorylated form, it would not change anything on our conclusion that ng_ζ1 drains precursors required for peptidoglycan synthesis.

Comment 13: Page 14: “...where transglycosylases catalyze the formation of alternating glycan chain strands” (insert underlined words), as the transglycosylases themselves do not

form the alternating glycan chain strands. What are the enzyme(s) that catalyze the formation of peptide bridges which lead to the formation of mesh-like architecture of the peptidoglycan layer?

Done. Peptide bridges are formed by transpeptidases. However, it is conceivable that these enzymes are only affected indirectly by ng_ζ1 as periplasmic peptidoglycan synthesis is affected by draining cytosolic processes and transglycosylation.

Comment 14: Page 15: “...reflect an adaptation of the zeta toxins to the peptidoglycan and cell wall structure of bacteria” (suggest addition of “and cell wall” as ng_ζ1 may affect the synthesis of the LPS layer of Gram-negative bacteria; “...very sensitive to changes...” (not “...sensitive on changes..”; final line page 15).

Done

Comment 14: Page 18: it was stated that for protein expression, *E. coli* BL21 (DE3)-RIL was used but subsequent experiments listed the use of *E. coli* C41(DE3) cells. Please provide the rationale for using this strain of *E. coli*.

When we tried to express toxic ng_ζ1 in *E. coli* BL21 (DE3)-RIL cells, we faced problems in plasmid stability or cultures even stopped growth spontaneously, a problem commonly known for toxin expression in this strain. However, we could overcome these problems by using the *E. coli* strain C41 (DE3), which according to literature and our experience is much more tolerant to expression of toxic proteins. As we were in the in the fortunate situation, that we could remove the antitoxin from the toxin during protein purification, we never had to express ng_ζ1 alone for any *in vitro* experiments. Just for historical reason, we used proteins purified from BL21 (DE3)-RIL expression as they had purities which were better than 99 %. We thank the referee for the comment and now included this argument in the material and method section. Furthermore, we want to apologize that we accidentally listed the strain BL21(DE3)-RIL in the legend for Fig. 1. This was incorrect, all these experiments were performed in C41(DE3) cells and the legend has been corrected.

Comment 15: Pages 20-21: synthesis and purification of UNAM, EP-UNAG, UNAM-4P and EPUNAG-4P. These UDP-derived compounds were synthesized by enzymatic conversion of

the relevant substrates. Were there any steps taken to validate the purified products? Please provide the relevant citation(s) for the ϵ_{260} value for the UDP-derived sugar species given in the last line of page 21.

We thank the referee for this comment. The identity of all products was validated by ESI-MS, and this information is now included in the manuscript. We also would like to point out, that UNAM and UNAG-EP was used for control experiments using MurB and MurC (Fig. 4) which are highly substrate specific. UNAM-4P was used for structure determination of the product bound state and UNAG-4P was characterized by NMR. Furthermore, phosphorylation by ng_ζ1 lead to significant tighter binding to the anionic exchange chromatography and thus to baseline separation of unphosphorylated and phosphorylated species, which we consider to be a standard prerequisite for such procedures.

Concerning the question of the extinction coefficient used in the manuscript: We have used the extinction coefficient for UDP as reported by Voet et al., Biopolymers, 1963 one of the standard values commonly used in the literature. According to Dwarson RMC (Data for biochemical research. Clarendon Press, Oxford 1986) we used the same extinction coefficient for all UDP-sugars. Both citations are now given in the manuscript.

Reviewer #2 (Remarks to the Author):

Comment 1: Supplementary Figure 1 and the whole concept of "shuffling" or rearrangement of the protein domains is not clear. I suggest improving the descriptions on p. 6 and 9 as well as redesigning the figure to make this clearer. I don't understand the grey lines in the figure and there is not much help in the legend.

We thank the reviewer for the comment and do understand that we were too short in the main text when describing the domain shuffling. When comparing the tertiary structures of the streptococcal toxins (we choose to show just the *S. pyogenes* homologue since it is structurally very similar to the pneumococcal PezT) an entire sequence block ranging from G59 to Q109 needs to be placed after K27 in order that the amino acid sequence of pyogenes ζ matches that of ng_ζ1. In fact, the grey lines which were poorly described in the legend, aimed at illustrating these connections to follow the progression within the amino acid sequence. We hope that this becomes now clearer from the improved figure legend. In conclusion, this causes that the tertiary structure is conserved except two connecting loops where the polypeptide chain connects to different regions within the progression of the

sequence. We do now better discuss this topological rewiring in the manuscript and hope to explain this sufficiently in the Supplemental Fig. 1 legend.

Comment 2: *Please also include an alignment for the antitoxin demonstrating that it is very divergent as claimed.*

Unfortunately, such an alignment would not be very helpful: The polypeptide chains differ substantially in length (61 aa in case of ng_ε1, 90 aa in case of streptococcal ε antitoxin), the sequence similarity is low (22.4 % homology) and the tertiary structure is very different not allowing to perform any structural alignment (see also comment 6 from reviewer 2 and the Supplemental Fig. 6).

Comment 3: *Page 8. Please include the single chain molecular masses for both toxin and antitoxin to allow the reader to assess the analysis of the complex stoichiometry.*

We are sorry that we just had mentioned the predicted molecular mass for the complex and not those of the individual polypeptide chains in the figure legend of Supplementary Fig. 5. The individual masses are now given as well.

Comment 4: *Page 8. I am missing a structural comparison to the heterotetrameric complexes, either in Fig. 3 or as a supplementary figure. I assume that the authors have checked the crystal packing carefully for a possible tetramer in this case?*

This is an excellent idea and we have now included Supplementary Fig. 6, which shows the superposition of ng_ε1 with ε from *S. pyogenes*. The superposition was performed by aligning the toxin kinase domain. The latter, however, is not shown as cartoon representation, as the figure would become too unclear. Nevertheless, it clearly shows that the antitoxins have different fold and dimerization similar to pyogenes ε is not possible in case of ng_ε1 (see comment 6 from reviewer 2). And yes, the crystal packing was checked carefully, and no additional extended contacts were observed. Furthermore, our SEC-MALS experiments shown in Supplemental Fig. 5 unambiguously corroborated the heterodimeric state in solution.

Comment 5: *Table 1. PDB entries missing.*

The structures have been deposited at the PDB and the entry numbers are included in Table 1.

Comment 6: *Figure 3a. I suggest you include an overview of the TA complex in cartoon representation as panel A. It is very difficult to understand the fold and mode of antitoxin interaction from the current figure with an electrostatic surface on the toxin part. And perhaps also a zoomed view of the TA interaction? I am also missing a comparison to the previously described, helical bundle antitoxins. Are there any structural similarity at all or are they completely different?*

We do now show the ng_ε1 antitoxin as ribbon representation also in Fig. 3b but would like to keep the surface representation in Fig. 3a for better visibility, how the antitoxin extends over the molecular surface of ng_ζ1. We think that a zoom in view of the interaction does not provide additional information, at least not more than already shown in Fig. 3b. Furthermore, we have included Supplemental Fig. 6 (see also comment 4 from reviewer 2) that shows how different the antitoxins are.

Comment 7: *Figure 3c. The legend says that the map is Fo-Fc but it is not clear to me if the ligand was omitted during refinement. In any case, you should show a complete unbiased electron density map calculated before the ligand was included in the refinement. Otherwise there is a slight risk of phase bias. I am also missing a structural comparison to the 3'OH reacting enzymes here to show why this toxin phosphorylates the 4'OH.*

We thank the reviewer for this excellent idea and have now included Supplemental Fig. 7 which shows a structural comparison of the active site of 4'-OH and 3'-OH phosphorylating enzymes. Secondly: Yes, the ligand was omitted and as the referee correctly pointed out, the electron density shown is a map before the ligand had been included for refinement. However, as this is standard of good scientific practice in the field, we had not mentioned it explicitly.

Comment 8: *Figure 3d. Labels for the sugar atoms (+phosphate) could improve the figure.*

We have labelled the 4'-phosphate group in figure 3d. However, labelling all sugar atoms would make it overcrowded and we feel that this is not necessary as the presence of the phosphate group at the C4' position, the lactoyl group on the C3' position and the N-acetyl

group at the C2' position of UNAM make of atom assignment anyway easy and unambiguous.

Comment 9: *Supplementary Figure 4. I don't understand why the chromatogram is printed so small when there is a whole page and this is a supplementary file. Please enlarge so one can see the light scattering data. The raw light scattering data should be shown as dots, not with a line.*

We apologize and have rescaled the chromatogram to fit the full page. Yet, the sampling rate of the MALS signal is still too small to be resolved into single dots. For clearer visibility, however, we would like to stick to the current spot size of individual measurements, although they might blur into a line.

Minor issues:

Page 6. This is perhaps pedantic, but I don't think you can question whether ng epsilon 1 is a "bona fide TA system at all" (end of paragraph), you can question whether it's a TA system and show that it's bona fide.

bona fide has been replaced by functional

Reviewer #3 (Remarks to the Author):

Comment 1: *The title should be more focused to better cover the main content of the manuscript. For example: "Zeta toxin of gonococci lyse bacteria by phosphorylation of multiple metabolites involved in cell wall synthesis" or something like that.*

We agree that the title was too generic and have changed it to:

"The ng_ζ1 toxin of the gonococcal Epsilon/Zeta System drains Precursors for Cell Wall Synthesis"

(See also reviewer #1 comment 1)

Comment 2: *Abstract, line 6: "zeta-related" is imprecise. They must mean "epsilon/zeta-homologous". The Zeta toxins exhibit primary, secondary and tertiary similarities and are*

therefore homologous. By inference, the modules/loci/systems must be homologous and not analogous.

We agree and have replaced the relatedness by homology.

Comment 3: *In that vein, it would be pertinent to include a supplementary Figure showing the genetic structure of the Gram-negative ϵ/\square modules and compare it to that of ϵ/\square modules of Gram positives.*

Similar to the majority of TA-operons, also the ng_ε1/ng_ζ1 locus is a bicistronic operon in which the stop codon of the antitoxin overlaps with the start codon of the toxin and is in this respect very similar to that of Gram-positive systems. Figures illustrating such bicistronic operons have been shown in a series of previous review articles on TA systems and additional such a figure has also been presented in the original paper by Pachulec & van der Does, PLoS one, 2010. We thus feel that such a figure would be redundant and not very helpful at that place. In case the reviewer asks about the genetic region surrounding the operon: The operon is located in a variable genetic load region in different conjugative plasmids (see response to comment 5 from reviewer 1) which already has been described in the original paper cited above and a finding that is now briefly discussed in the manuscript.

Comment 4: *P3, line 7: It would be fair here to include references to the discovery of ribosome-dependent and independent TA-encoded toxins/mRNases.*

We have now also included references for the original manuscript from Pedersen et al. and Zhang et al. both published in 2003, which describe the first direct evidence that either RelE or MazF are ribonucleases.

Comment 5: *P3, line 9: same with reference 17, replace it or include a reference to the discovery of membrane-damaging toxins.*

We have now included a reference for the pivotal study in this field showing that the hok gene product integrates into the membrane contributed by Kenn Gerdes et al. published in 1986.

Comment 6: P3, line 13 and other place: Replace “genuine” with “canonical” or “bona fide”

Done (2x).

Comment 7: P4, line 1: unclear what “zeta-dormant” means. Probably the authors means when Zeta toxin is inactive (due to being in complex with Epsilon). Please amend.

We changed this to: ...When zeta is inactive, e.g. by binding to the epsilon antitoxin,...

Comment 8: Supp. Fig. 1: It would be useful if the authors could add a phylogenetic tree of Zeta toxins, indicating the organisms in which the toxins are present.

We have included a Supplemental Table 1 in which the currently identified homologous proteins of ng_ζ1 are given (see also our response to comment 9 from reviewer 1). We fully agree with the reviewer, that the evolutionary history of the system would be interesting, which, however, will not become amenable from a simple phylogenetic tree. The ng_ε1/ng_ζ1 systems are located on plasmids (see response to comment 5 from reviewer 1) and have apparently been acquired by horizontal gene transfer and the spread of a certain operon is also determined by the ability of the entire plasmid to spread over different organisms and being maintained. Moreover, we think a useful phylogenetic tree should also include genetic information (such as codon variations and polymorphism etc.) and would require a thorough bioinformatical analysis that would be far beyond the scope of the manuscript.

Comment 9: P5, line 10: delete “at all”

Done

Comment 10: P8: consider moving Table 1 to Table S1

The crystallographic Table 1 contains all important information required to judge the quality of the structural part of the manuscript and thus, we would like to keep it in the manuscript: a practice that is still considered as being the gold standard in the X-ray crystallography community.

Comment 11: *P11: The section starting with “A non-discriminative....” must be presented in a clearer way. One possibility is to add an overview Figure showing that pathway of Cell Wall synthesis, showing intermediates and enzymes, and indicating which metabolite intermediates are phosphorylated by Zeta toxins from Gram-negative and –positive bacteria. Since the manuscript has only 4 main Figures there is ample space for an additional Figure.*

This is an excellent idea and we have now included an additional figure (Fig. 5) which we introduce at the end of the paragraph, where the results are briefly recapitulated. Furthermore, we have revised the beginning of the section in hope that the conundrum posed from the different enzymatic properties of ng_ζ1 observed *in vitro* and the accumulating products observed *in vivo* (see also our reply to comment 12 from reviewer 1) becomes clearer to the reader and better explains why the reported experiments are of fundamental importance for the presented model.

Comment 12: *Consider shortening of the Discussion.*

We reconsidered our discussion did not find any strong redundancy. Thus, we feel that we have to disagree with the reviewer and would like to keep the discussion in the current format.

REVIEWERS' COMMENTS:

Reviewer #1 (Remarks to the Author):

The revised manuscript by Rocker et al. is substantially improved and the authors have addressed all my queries in my review report. Thus, I am of the opinion that the manuscript is now acceptable for publication in Nature Communications.

I only found very minor typographical errors in the revised manuscript:

Supplementary file, line 118: Sentence should perhaps be "Conserved residues mapped on the ζ sequence from..." (instead of "Conservation mapped on the sequence ζ from..."); line 128: "...both sides of the connecting clamb"? (should be "connecting clamp"?)

We are grateful the reviewer for the comment, which we have implemented in the revised version of the manuscript.

Reviewer #1 (Remarks to the Author):

Comment 1: The revised manuscript by Rocker et al. is substantially improved and the authors have addressed all my queries in my review report. Thus, I am of the opinion that the manuscript is now acceptable for publication in Nature Communications.

I only found very minor typographical errors in the revised manuscript:

Supplementary file, line 118: Sentence should perhaps be "Conserved residues mapped on the ζ sequence from..." (instead of "Conservation maped on the sequence ζ from..."); line 128:

"...both sides of the connecting clamb"? (should be "connecting clamp"?)

We want to apologize for these mistakes and have corrected the Supplementary information accordingly.